# Fully automatic image colorization based on semantic segmentation technology

**Min Xu, YouDong Ding***

Shanghai Film Academy, Shanghai University, Shanghai, PR China

* ydding@shu.edu.cn, shudu17@foxmail.com

**Data Availability Statement:** The images used are from the ILSVRC2012 dataset (http://www.image-net.org/challenges/LSVRC/2012/index) and the Avopix (https://avopix.com/). All other relevant data

## Abstract

Aiming at these problems of image colorization algorithms based on deep learning, such as color bleeding and insufficient color, this paper converts the study of image colorization to the optimization of image semantic segmentation, and proposes a fully automatic image colorization model based on semantic segmentation technology. Firstly, we use the encoder as the local feature extraction network and use VGG-16 as the global feature extraction network. These two parts do not interfere with each other, but they share the low-level feature. Then, the first fusion module is constructed to merge local features and global features, and the fusion results are input into semantic segmentation network and color prediction network respectively. Finally, the color prediction network obtains the semantic segmentation information of the image through the second fusion module, and predicts the chrominance of the image based on it. Through several sets of experiments, it is proved that the performance of our model becomes stronger and stronger under the nourishment of the data. Even in some complex scenes, our model can predict reasonable colors and color correctly, and the output effect is very real and natural.

## Introduction

Many fields, including old photo and old movie restoration, remote sensing image, and biologic medical image, have strong demand for image colorization technology. The goal of image colorization is to assign colors to each pixel of a grayscale image, and the researches on this subject has also been in the ascendant. The earliest research on this subject was Markle [1], who realized the colorization of the moon image with the help of computer aided technologies, which attracted wide attention from all walks of life. In the past, most commonly used methods for processing image colorization were extension method based on local color [2, 3] and color transfer method based on the reference image [4, 5]. The biggest advantage of the former is reflected in the interactivity and controllability. The user can color the target image according to their own will, and it can get better coloring effect even for target images with complex content. The disadvantage is that the algorithm has certain requirements on the user's own color sensitivity and color matching. In addition, it is prone to problems such as color bleeding and boundary blurring when dealing with images with complex textures. Therefore, it is only suitable for application scenarios with low requirements on the accuracy of the colorization of the

are within the manuscript and its "Supporting information".

**Funding:** This project was funded by National Natural Science Foundation of China under the grant 61303093 and 61402278. The funders had no role in study design, data collection and analysis, decision to publish, or preparation of the manuscript. None of the authors received salaries from any of the funders.

**Competing interests:** The authors have declared that no competing interests exist.

border. The advantage of the latter is that the influence of human factors is eliminated in the coloring process, and the result is relatively objective. The limitation is that the coloring effect is completely dependent on the similarity between the reference image and the target image, so it is only suitable for image colorization with a single hue or content.

Deep neural networks realize image colorization that have gradually became a trend to replace manual coloring now. Compared with the above two methods, this end-to-end network overcomes the limitation of human intervention, and it is natural, efficient and easy to operate. The goal of this paper is to convert the grayscale image or the color image without rich information into the color image with clear details and clear colors, so as to improve the visual effect of users and facilitate the study of subsequent images. In theory, different landscape's colors in black-and-white images have different grayscales, so neural network based on grey values of images can judge color of the item roughly. The result may not be very accurate, especially the pixels of similar grayscales, such as grass may become blue and blue jeans may became green through the calculation of neural network. In addition, color bleeding is also a common problem in image colorization. Therefore, we need to let neural networks have common sense so that it can judge where the boundaries of objects are and what colors the items should be in different scenarios. Aiming at the above two problems, this paper proposes a fully automatic grayscale image colorization model based on semantic segmentation technology, and its colorization process is shown in Fig 1.

The contributions of this paper include: (1) histogram equalization effectively improves the visual effect and the colorfulness of overexposed and underexposed images; (2) the introduction of semantic segmentation network accelerates the edge convergence of the image and improves the positioning accuracy of the algorithm, and solves the problem of color bleeding;

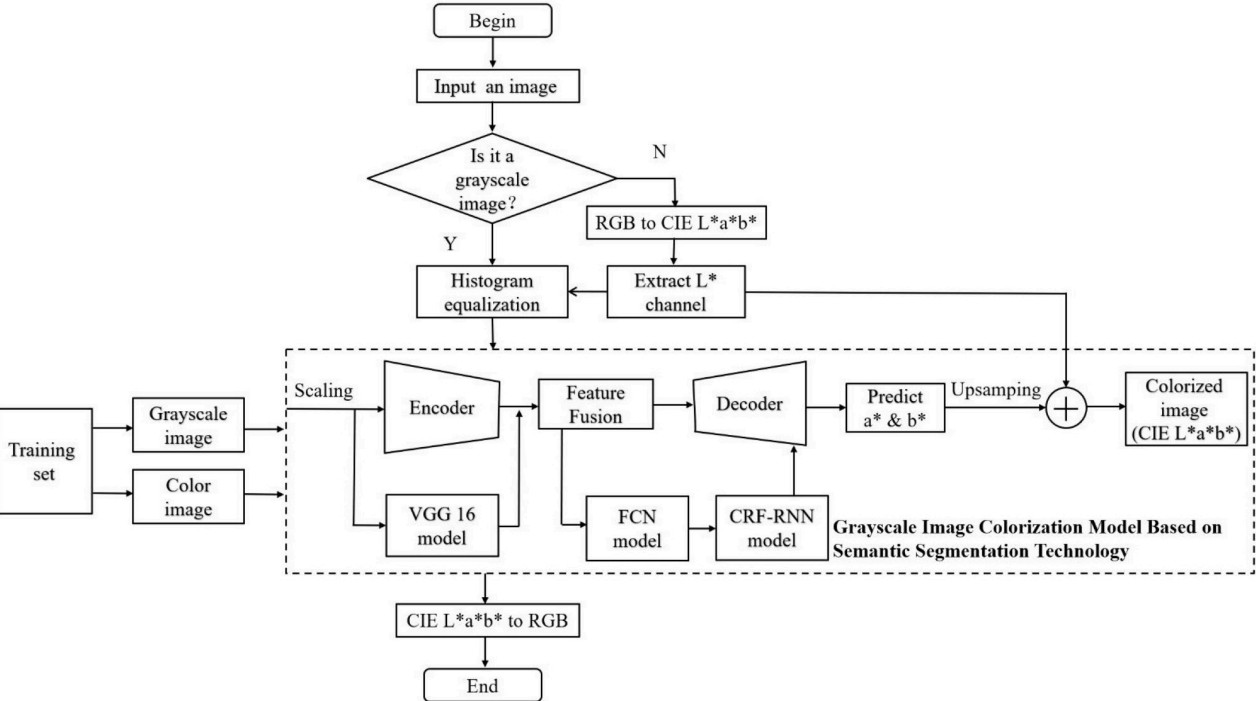

**Fig 1. Colorization process of our algorithm.** In the dotted box, the corrected grayscale image is fed into the network model for color prediction of $a^*$ and $b^*$.

(3) compared with several popular algorithms, our model has better results in natural images colorization and black-and-white images colorization.

## Related work

The neural network models currently used for image colorization are mostly generative adversarial networks(GAN) [6] and convolutional neural networks(CNN) [7].

### Research on image colorization based on GAN

In recent years, GAN has achieved great success in the fields of image generation, image translation and image restoration. Isola et al. [8] proposed pix2pix that realized the image transform of two types, which basically solved the problem of grayscale images colorization, but its disadvantage was that it was difficult to obtain paired data for training in real life. Zhu et al. [9] presented CycleGAN on the basis of pix2pix, which introduced the reconstructed loss to achieve the separation of style and content. CycleGAN supports the training of unpaired data to achieve one-to-one style transfer and image colorization, but the coloring object must be the object that appears in the training set, otherwise the incorrect color may be passed to it. Zhu et al. [10] put forward BicycleGAN, which combined the advantages of VAE GAN [11] and LR GAN [12]. BicycleGAN can be regarded as the upgraded version of pix2pix, which requires paired data to train and has features of parameter sharing in CycleGAN. These models are more accurate in the treatment of highly recognizable colors such as portraits, plants, and the sky. For the difficult part of identification, more warm colors are used or the contrast is simply enhanced.

### Research on image colorization based on CNN

CNN models that are similar to AlexNet [13] and VGGNet [14] are usually used for image classification and regression tasks, while image colorization can be regarded as the prediction of the probability of color of each pixel of grayscale image, which is similar to regression tasks. Cheng et al. [15] used CNN to extract the high-level features of the image, realized automatic image colorization by inputting the image descriptors. Iizuka et al. [16] constructed a fusion layer to fuse the local image block information with the global prior of the whole image to predict the color information of each pixel in the grayscale image, but the coloring result was fixed. Larsson et al. [17] used the natural appearance of multimodal color distribution of the image scenes to train the model for generating corresponding color histogram and color image. The effect of their approach was better than others. Zhang et al. [18] were inspired by "color prediction is a multimodal problem" [14], and predicted the color distribution for each pixel, so that the final result could be in a variety of different styles. Zhang et al. [19] introduced an AI tool for real-time coloring black-and-white images by fusing low-level clues and high-level semantic information, which could directly map grayscale images to CNN to generate colorized results. He et al. [20] proposed an example-based deep learning method for local colorization, this network allowed different reference images to be selected to control the output image. Even if unrelated reference images are used, this method has good robustness and versatility.

### Other methods

In addition to image content, color also affects the user's emotional response. Yang et al. [21, 22] used color histogram to convey emotions between color images. This method is very flexible and supports users to choose different references to convey different emotions. Wang et al.

[23, 24] obtained a sentiment palette by giving a semantic word or providing a reference image, and directly transferred the color of the template to the target image. However, these studies [25–28] incorporated emotional factors they had ignored into the image. Cho et al. [25, 26] proposed Text2Colors that this model included text-to-palette generation networks and palette-based colorization networks, these two sub-networks utilized conditional GAN (cGAN) [12]. Chen et al. [27] used the recurrent attention model (RAM) to fuse images and semantic features, introduced a stop gate for each region of the image, so as to dynamically determine whether to continue to infer additional information from the text description after each reasoning step, and finally showed the first semantic based coloring result on the Oxford-102 Flowers dataset [28]. Su and Sun [29] improved the previous idea of using a single color or several colors to achieve color transform, and proposed that the user could adjust the number of main colors in the image according to the complexity of the image content. The operation is more flexible and the coloring effect is more accurate and natural. Wan S et al. [30] input the extracted points of interest into the network to generate color points, and continued to spread to neighboring pixels to achieve fully automatic image coloring. Through several sets of comparative experiments, it can be seen that this method is very efficient and has a good application prospect. At the same time, the author also talked about the subsequent application of the model to low-light night vision images in the conclusion section, which will usher in greater challenges. Yu X et al. [31] first confirmed the scene category of the input image, and then performed color mapping learning based on the image in the corresponding category, which greatly improved the coloring efficiency and coloring accuracy of the algorithm. The idea of guiding coloring according to the use scene is very targeted and very suitable for coloring medical images. Special applications in other fields can also be considered in the future.

## Proposed method

We borrowed the architecture design of Iizuka et al. [16] on the feature extraction network, and combined the actual needs to introduce the histogram equalization and semantic segmentation technology, which makes the final coloring effect very good. The model mainly includes six parts: low-level feature extraction network, local feature extraction network, global feature extraction network, the fusion modules, semantic segmentation network, and color prediction network. The main line of the network is a U-Net [32] with an encoding-decoding structure, the specific design is shown in Fig 2. The encoder allows us to input images of arbitrary

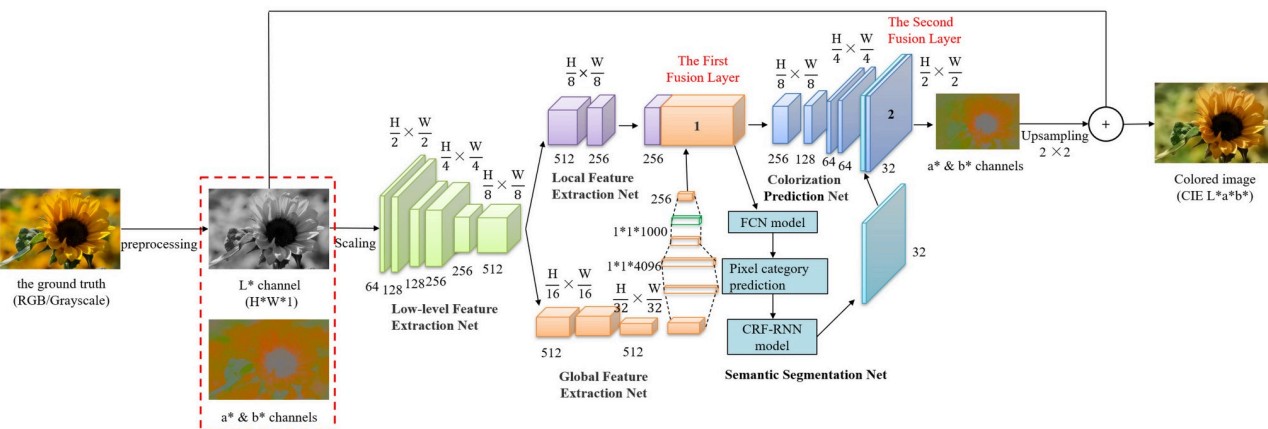

**Fig 2. Our model.** Some semantic segmentation information of the image can help the prediction network understand the content of the image and coloring accurately.

resolution. The core of the color prediction network is the decoder, which is responsible for predicting a* and b* of the image based on the output of the encoder and the learning effect of the semantic segmentation network.

### Feature extraction network

Our feature extraction network includes low-level feature extraction network, local feature extraction network and global feature extraction network. In order to reduce the difficulty of network training, the encoder on the left of U-Net is selected as the local feature extraction network, and VGG-16 [14] is used to extract the global feature, and they share the low-level feature.

**Low-level feature extraction network.** Before entering the low-level feature extraction network, the target image needs to be normalized to a size of $224 \times 224$, because the size of the fully connected (FC) layer needs to be fixed to complete feature fusion and semantic classification in the following global feature extraction and semantic segmentation network. The low-level feature extraction network uses six convolutional layers to extract the low-level features, and Eq (1) is as follows:

$$
\begin{cases}
Y_{u,v}^{p,q} = \delta\left(\sum_{i=-k_h'}^{k_h'} \sum_{j=k_w'}^{k_w'} W_{k_h'+i,k_w'+j}^{p} \cdot X_{u+i,v+j}^{q} + b\right), \\
k_h' = \dfrac{k_h - 1}{2}, \\
x_w' = \dfrac{k_w - 1}{2}.
\end{cases}
\tag{1}
$$

Where $(u, v)$ is the coordinate of a pixel, $k_h$ and $k_w$ are the height and the width of the convolution kernel. $X_{u,v}$ and $Y_{u,v}$ are the values of the input and the output under the coordinates $(u, v)$. $W \in R^n$ is the weight matrix of the 2D convolution kernel, $b$ is the bias term, $\delta$ is a non-linear activation function, $p$ is the number of convolution kernels, $q$ is the number of channels of the input unit. The Rectified Linear Unit (ReLU) is used to complete the convolution calculation, Eq (2) is as follows:

$$
\delta_{ReLU}(x) = max(0, x).
\tag{2}
$$

**Local feature extraction network.** After obtaining the low-level features of the image, the model is initially divided into two branches, one of which uses two convolutional layers to calculate local semantic features, and it obtains the local features. Here, $H$ and $W$ are the height and width of the input image.

**Global feature extraction network.** At the same time, another branch processes the low level features by additional four convolutional layers and three FC layers to get the global features, a 256 bins vector.

In order to retain as many semantic features of the image as possible, we did not pool the features after convolution, but increase the step size of the convolution kernel to achieve pooling effect and reduce the dimension of the features correspondingly. As a result, not only the semantic features would be retained, but also the feature size, the noise, and the parameters number would be reduced.

## Two fusion modules

In order to get better coloring results, this paper introduces two fusion modules. The first fusion layer is responsible for fusing global features and local features together. The convolution kernel size of this network is $1 \times 1$, and the step size is 1. The second fusion layer is responsible for combining the image's semantic segmentation information when predicting the color of the image, which can color more accurately and prevent color bleeding.

## Semantic segmentation network

FCN [33] is regarded as a pioneering work in the field of image semantic segmentation, and many semantic segmentation models have been proposed based on FCN. For example, some models improve the structure of the network (SegNet [34], DeconvNet [35]), and some models improve the convolution kernel (DeepLab [36]), the most important is the introduction of markov random field (MRF) on the basis of the rough semantic map to smooth the segmentation of edges [37]. This paper makes full use of the local and global features of the image, and uses the FCN model first to segment the target image into categories such as plants, buildings, sky, water, roads and etc., then calculates the color of each category, and finally calculates the hue value of each block by using a probabilistic method to mix the feature color of each category. The data from a particular paper [38] shows that the addition of conditional random field (CRF) can improve the final score by $1 - 2\%$.

Given that image data is the set of observable variables $X = \{X_i\}_{i=1}^N$, and the set of hidden variables to be inferred is $Y = \{Y_i\}_{i=1}^N$, both are sequences of random variables are by linear chains. According to the conditional probability model $P(Y|X)$ proposed in this paper [39] to predict the label of each pixel, and it satisfies markov process. Eq (3) is as follows:

$$P(Y_i|X, Y_1, ..., Y_{i-1}, Y_{i+1}, \ldots, Y_N) = P(Y_i|X, Y_{i-1}, Y_{i+1}).  \tag{3}$$

Where $Y$ is the tag sequence or state sequence of the output, and its value is the category label $\{1, 2, \ldots, L\}$. So the output of FCN is a $L$ bins vector, where each bin represents the probability that the set of hidden variables belongs to this class.

According to hammersley-clifford theorem, the factorization formula of linear chain random field $P(Y|X)$ can be given, and each factor is a function defined at two adjacent nodes. Given the condition is x, y, Eq (3) could be written as Eq (4):

$$P(y|x) = \frac{1}{Z(x)} exp \left\{ \sum_{i,m} \lambda_m t_m(y_{i-1}, y_i, x, i) + \sum_{i,n} \mu_n s_n(y_i, x, i) \right\}.  \tag{4}$$

Where $Z(x)$ is the normalized term, whose sum is carried out on all possible output sequences. Eq (5) is as follows:

$$Z(x) = \sum_y \left\{ \sum_{i,m} \lambda_m t_m(y_{i-1}, y_i, x, i) + \sum_{i,n} \mu_n s_n(y_i, x, i) \right\}.  \tag{5}$$

Here, $t_m$ and $s_n$ are characteristic functions, $\lambda_m$ and $\mu_n$ are corresponding weights. $t_m$ is a feature function defined on an edge, called a transition feature, and it depends on the current and previous positions. $s_n$ is a feature function defined on a node, called a state feature, and it depends on the current position. In general, the local feature functions $t_m$ and $s_n$ take values of 1 or 0.

Given $K_1$ transition features and $K_2$ state features, then we can unify the feature as Eq (6).

$$f_k(y_{i-1}, y_i, x, i) = \begin{cases} t_m(y_{i-1}, y_i, x, i) & k = 1, 2, \ldots, K_1 \\ s_n(y_i, x, i) & k = K_1 + l; l = 1, 2, \ldots, K_2 \end{cases} \tag{6}$$

Next, the transition feature and the state feature are summed at various positions $i$, and it can be expressed as Eq (7).

$$f_k(y, x) = \sum_{i=1}^{n} f_k(y_{i-1}, y_i, x, i) \qquad k = 1, 2, \ldots, K_1 + K_2 \tag{7}$$

The weight $W_k$ corresponding to $f_k(y, x)$ is shown in Eq (8).

$$W_k = \begin{cases} \lambda_m & k = 1, 2, \ldots, K_1 \\ \mu_n & k = K_1 + l; l = 1, 2, \ldots, K_2 \end{cases} \tag{8}$$

Therefore, the CRF can be expressed as Eq (9).

$$P(y|x) = \frac{exp\{\sum_{k=1}^{K_1+K_2} W_k f_k(y, x)\}}{\sum_y exp\{\sum_{k=1}^{K_1+K_2} W_k f_k(y, x)\}}. \tag{9}$$

The parameter setting of the convolutional layer of the FCN is the same as that of the left side of U-Net. The difference is that it adds two convolutional layers, three FC layers and Softmax function after the first fusion module. To remove the spatial information and train the model to output a scalar, the result of our classification, the original 2D vector is converted to a 1D vector. These parameter settings of semantic segmentation network are shown in Table 1, Softmax function is as follows:

$$\delta_{Softmax}(i) = \frac{e^i}{\sum e^j}. \tag{10}$$

## Color prediction network

Color prediction network is to predict a* and b* according to the feature tensor and semantic segmentation information of the input image. Its core is the decoder on the right side of the U-Net, which is composed of the convolution layer and the upsample layer. The output image tensor is required to be $H \times W \times 2$, and these parameter settings are shown in Table 2.

The convolutional layer cuts down the information of the image, so the image proportion can be kept constant by adding blank padding. The upsample layer can double the resolution of the image, and if the two are used together, they not only can increase the information density, but also can't distort the image. To compare the difference between the predicted value

Table 1. Parameter settings of semantic segmentation network.

| Layer | Kernels | Stride | Output |
|-------|---------|--------|--------|
| conv | $3 \times 3$ | $2 \times 2$ | 512 |
| conv | $3 \times 3$ | $1 \times 1$ | 512 |
| FC | - | - | 1024 |
| FC | - | - | 256 |
| FC | - | - | 2 |

**Table 2. Parameter settings of color prediction network.**

| Layer | Kernels | Stride | Output |
|-------|---------|--------|--------|
| conv | $3 \times 3$ | $2 \times 2$ | 256 |
| conv | $3 \times 3$ | $1 \times 1$ | 128 |
| upsamp | - | - | 128 |
| conv | $3 \times 3$ | $1 \times 1$ | 64 |
| conv | $3 \times 3$ | $1 \times 1$ | 64 |
| upsamp | - | - | 64 |
| conv | $3 \times 3$ | $1 \times 1$ | 32 |
| conv | $3 \times 3$ | $1 \times 1$ | 2 |
| upsamp | - | - | |

and the actual value, we use the Tanh function, because the input of the Tanh function can be any value and the output is [−1, 1]. The Tanh function is as follows:

$$\delta_{Tanh}(x) = \frac{\sinh x}{\cosh x} = \frac{e^x - e^{-x}}{e^x + e^{-x}}. \tag{11}$$

Since the color values of a*b* are distributed in the interval [−128, 128], it is necessary to divide all values of the output layer by 128 to ensure that these values are in the −1 to 1 range for the convenience of comparing the errors of the predicted results. After the final error is obtained, the network will update the filter to reduce the global error, and improve the feature extraction effect through back propagation based on the error of each pixel until the error is small enough. After the neural network is trained, all the results are multiplied by 128 and converted back to the CIE L*a*b* image for the final prediction. There is no direct conversion between RGB colorspace and CIE L*a*b* colorspace, but it exists a XYZ colorspace that can help convert the two to each other: RGB ↔ XYZ ↔ CIE L*a*b*.

## Objective function and network training

In this paper, the loss of the model includes the loss of color prediction and the loss of semantic segmentation.

To quantify the loss of the model, we calculate the mean square error between the estimated pixel color in a*b* colorspace and the actual value. For image $X$, the loss of its color prediction network is as follows:

$$L(y^{color}) = C(X, \theta) = \frac{1}{2HW} \sum_{k \in \{a,b\}} \sum_{i=1}^{H} \sum_{j=1}^{W} (X_{k_{i,j}} - \tilde{X}_{k_{i,j}})^2. \tag{12}$$

Where $\theta$ are the parameters of all models, $X_{k_{i,j}}$ and $\tilde{X}_{k_{i,j}}$ are the $i_{th}$ and $j_{th}$ pixel values of the $K_{th}$ component of the target image and the reconstructed image respectively.

Semantic segmentation network can help color prediction network to learn how to supplement color information, so it is necessary to calculate the loss of semantic segmentation. In this paper, the loss function of semantic segmentation is defined as Eq (13).

$$L(y^{class}) = -\sum_{H,W} V_s(X_{H,W}) \log(E(X_{H,W}; \theta)). \tag{13}$$

Where $V_s$ is the weight of rebalancing losses. The total loss of this network can be expressed as

Eq (14).

$$L_G(y^{color}, y^{class}) = \eta_1 L(y^{color}) + \eta_2 L(y^{class}). \tag{14}$$

Where $L(y^{class})$ is the loss of semantic segmentation network, $L(y^{color})$ is the loss of color prediction network, $\eta_1$ and $\eta_2$ are the correlation of the two losses.

## Experiments

### Experimental environment and dataset

All tests are run on an NVIDIA GTX 1080 TITAN GPU. According to the hardware, we divided the training 2000000 images into 15625 batches with batch size is 128. In addition, Adam [40] optimization algorithm is also used to accelerate the training speed.

All training images and validation images in our model and several comparison algorithms in this paper are from the same public dataset, ILSVRC2012 [41], which is the dataset of the famous ImageNet [42]. All test images shown in the manuscript could be obtained from the support information (S2 File).

### Performance evaluation index

This paper uses image colourfulness(C), the peak signal-to-noise ratio (PSNR), the structural similarity (SSIM), the quaternion structural similarity (QSSIM) and the qualitative evaluation by user study to evaluate the performance of these algorithms. We will use the colorfulness metric methodology described in Hasler's paper [43], Eq (15) is as follows:

$$\begin{cases} rg = R - G, \\ yb = (R+G)/2 - B, \\ \sigma_{rgyb} = \sqrt{\sigma_{rg}^2 + \sigma_{yb}^2}, \\ \mu_{rgyb} = \sqrt{\mu_{rg}^2 + \mu_{yb}^2}, \\ C = \sigma_{rgyb} + 0.3 \times \mu_{rgyb}. \end{cases} \tag{15}$$

Where $\sigma_{rg}$ is the standard of $rg$ and $\mu_{rg}$ is the deviation of $rg$, as $\sigma_{yb}$ and $\mu_{yb}$ are to $yb$. $C$ is described the colorfulness of the image.

PSNR is obtained from the mean square error (MSE), and it is defined as follows:

$$PSNR = 10 \times \log_{10}\left(\frac{MAX_I^2}{MSE}\right). \tag{16}$$

$$MSE = \frac{1}{mn} \sum_{i=0}^{m-1} \sum_{j=0}^{n-1} \|I(i,j) - K(i,j)\|^2. \tag{17}$$

Here, $MAX$ refers to the grayscale of the image, which is generally 255. $MSE$ is the mean square error between the original image $I$ and the processing image $K$. $m$ and $n$ are the number of rows and columns of the images respectively.

SSIM uses the mean value as the estimation of the luminance $L$, the standard deviation is estimated as the contrast degree $C$, and the covariance is estimated as the structural similarity

degree $S$, and the mathematical model is calculated as follows [44]:

$$\begin{cases} L(x,y) = \dfrac{2\mu_x\mu_y + c_1}{\mu_x^2 + \mu_y^2 + c_1}, \\[2mm] C(x,y) = \dfrac{2\sigma_x\sigma_y + c_2}{\sigma_x^2 + \sigma_y^2 + c_2}, \\[2mm] S(x,y) = \dfrac{\sigma_{xy} + c_3}{\sigma_x\sigma_y + c_3}, \\[2mm] SSIM(x,y) = L(x,y)^\alpha \times C(x,y)^\beta \times S(x,y)^\gamma. \end{cases} \qquad (18)$$

Here, $\mu_x$ and $\mu_y$ are the mean of image $x$ and $y$ respectively, $\sigma_x$ and $\sigma_y$ are the variance of image $x$ and $y$ respectively, $\sigma_{xy}$ is the covariance of image $x$ and $y$. To avoid having a zero in the denominator, we introduce $c_1$, $c_2$, $c_3$. We usually set: $c_1 = (K_1 \times L)^2$, $c2 = (K_2 \times L)^2$, $c_3 = c_2/2$, $K_1 = 0.01$, $K_2 = 0.03$, $L = 255$. Eq (19) is rewritten as follows:

$$SSIM(x,y) = \frac{(2\mu_x\mu_y + c_1)(2\sigma_{xy} + c_2)}{(\mu_x^2 + \mu_y^2 + c_1)(\sigma_x^2 + \sigma_y^2 + c_2)}. \qquad (19)$$

PSNR and SSIM are the most common and widely used full-reference image quality evaluation indexes. The PSNR value cannot well reflect the subjective feelings of the human eye. The calculation of SSIM is a little complicated, and it circumvents the complexity of natural image content and the problem of multi-channel uncorrelation to some extent, and measures the similarity of two images by directly estimating the structural changes of two complex structural signals. Its value can better reflect the subjective perception of the human eye, but it is only suitable for measuring the structural similarity between grayscale images. QSSIM is a new color image quality index, and SSIM can also be regarded as a special case of QSSIM [45]. Eq (20) is as follows [46]:

$$QSSIM_{ref,deg} = \left| \left( \frac{2\mu_{q_{ref}} \cdot \mu_{q_{deg}}}{\mu_{q_{ref}}^2 + \mu_{q_{deg}}^2} \right) \left( \frac{\sigma_{q_{ref},deg}}{\sigma_{q_{ref}}^2 + \sigma_{q_{deg}}^2} \right) \right|. \qquad (20)$$

Note: for the definition and value of parameters in Eq (20), please refer to reference [46].

## Experimental results and analysis

Before training the network, the RGB colorspace of the image is converted to the CIE L*a*b* colorspace. During training the network, the learning rate is initialized to 0.001, momentum is initialized to 0.5, and weight decay is initialized to 0.0005. After training the network, all results are multiplied by 128 to convert back to the RGB image.

**Comparison of coloring effects under different epochs.** Fig 3(a) shows the comparison of the coloring effects of our model on eight images after the training of the 10th, 20th, 30th, 40th and 50th generations. Fig 3(b)–3(g) are the magnification effect of two groups of images in Fig 3(a). Through comparing the coloring effects of the five epochs, we find that with the increase of the number of epoch, the number of weight updating iterations in the neural network increases, the color bleeding decreases, and the coloring effect of the image is closer and closer to the ground truth.

To further verify that the image quality is affected by the epoch size, we use line graphs to show the change of the PSNR values, the SSIM values and the QSSIM values of the above eight groups of images under different epochs, as described in Fig 4. In the eight groups of images

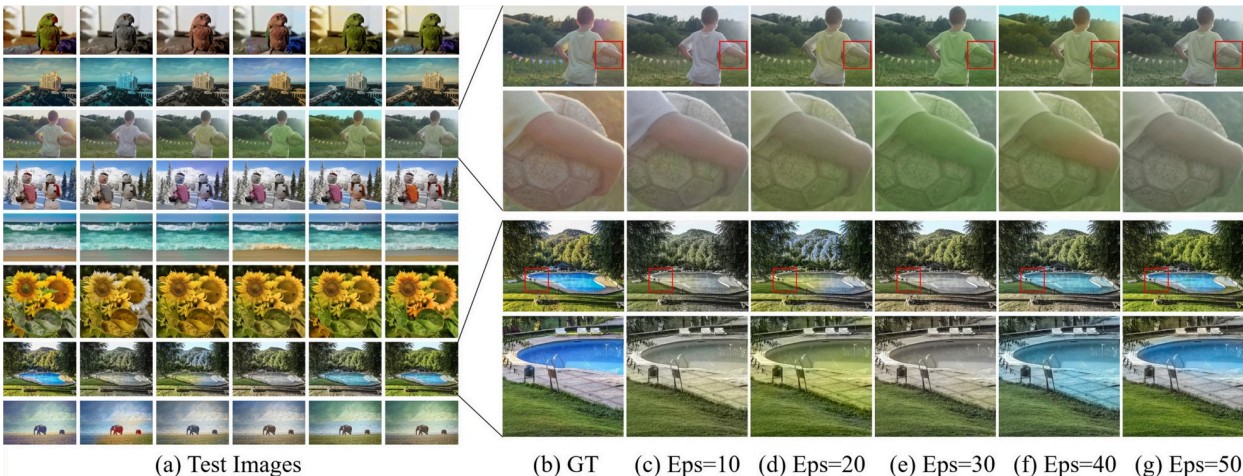

(a) Test Images (b) GT (c) Eps=10 (d) Eps=20 (e) Eps=30 (f) Eps=40 (g) Eps=50

**Fig 3. Comparison of coloring effects under different epochs.** The full name of GT is the ground truth, that is, is also the source image. The full name of Eps is Epochs. For convenience, the following image subtitles will use GT instead of the ground truth.

given, with the increase of the number of epoch, the PSNR values of half of the images have been changing and the corresponding curve fluctuates greatly, up to $13.471dB$. The PSNR values of the other half of the images are relatively stable, and the two extremes fluctuate within the range of about $5dB$. On the whole, the PSNR values of the eight groups of images are all above $29dB$ and tend to be stable when the epoch is 40. The SSIM values of the images are very stable overall, and the fluctuation range of each image does not exceed 0.01. The QSSIM values of the images are also relatively stable on the whole, and the fluctuation range is less than 0.02. In other words, the increase in the number of epoch would affect the PSNR value of some images, but basically would not affect the SSIM value and QSSIM value of images.

**Effect of histogram equalization on image colorization.** Theoretically, histogram equalization will result in the merge of gray levels, which may reduce the colorfulness of the image. It is found that histogram equalization in advance can eliminate the clutter and enhance the contrast of the image in the colorization study of some overexposed or underexposed images, and it can increase the colorfulness of the image in the later coloring process. Fig 5 is the image and its histogram before and after gray histogram equalization. It can be seen that the probability density of the gray level of the transformed image is evenly distributed, and the brightness and contrast of the whole image also become relatively natural. The pre-processed flower has clearer details and edges, which is very helpful in improving the accuracy of the model's color prediction.

Iizuka et al. [16] is the object for reference in this paper, which is also one of the most classical algorithms that applied deep learning to color prediction and achieved good coloring effects. Moreover, it does not consider coloring underexposed or overexposed images, so it will be selected as a comparative document in this section. For a specific comparison of visual sensory effects, please see Fig 6. From the horizontal perspective, histogram equalization can remove some clutter, save super dark images (underexposed images) and super bright images (overexposed images), and thus affect the final texture and color. For example, for the images with normal exposure in the first two lines in Fig 6, whether to add histogram equalization into our model has little influence on the final coloring effect, which is also the case in literature [16]. For the images with abnormal exposure in the last four lines, the coloring effect of

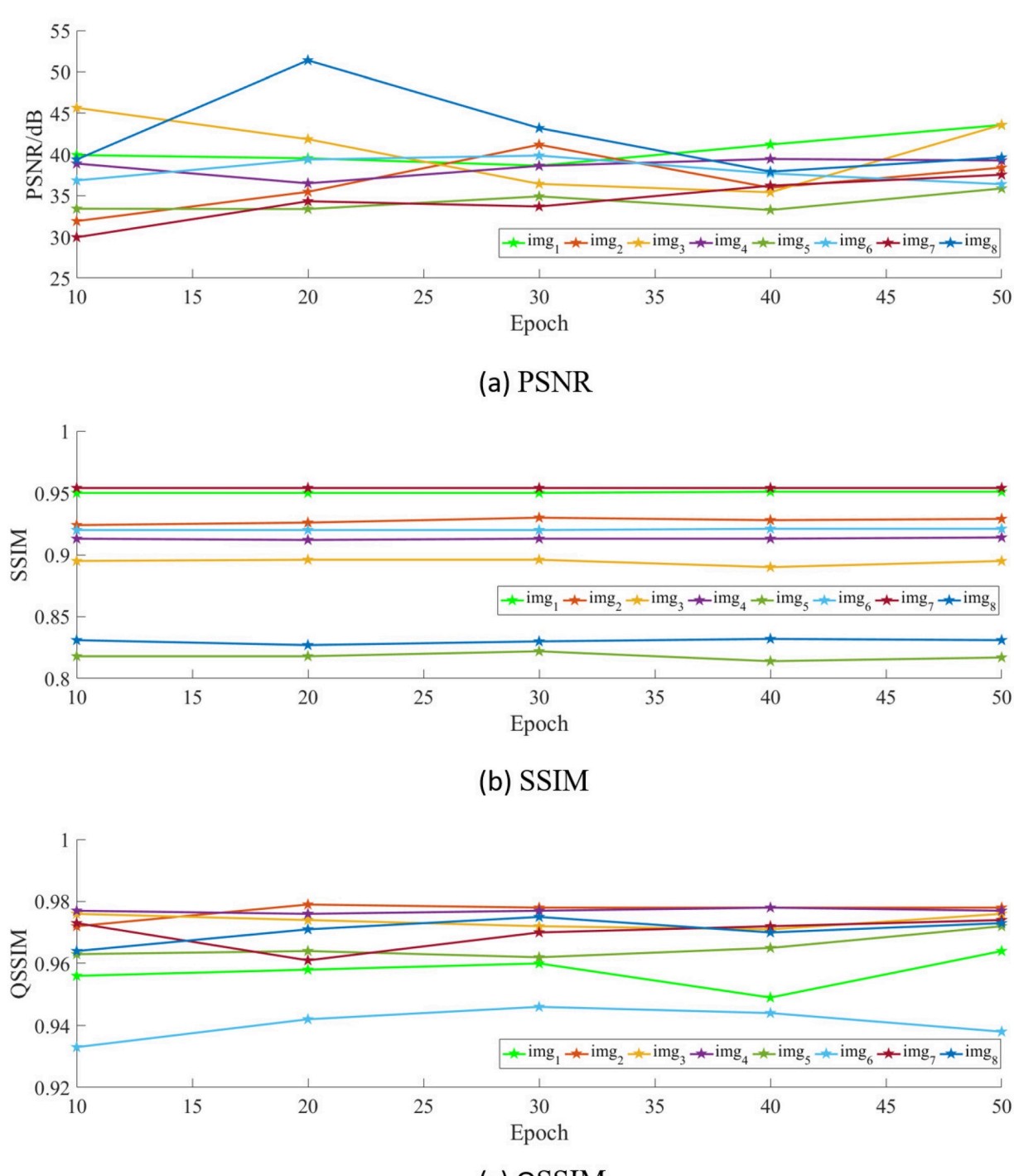

**Fig 4. Quantitative evaluation.** The data in this figure is from Fig 3, and the images are in the same order.

our model with histogram equalization is obviously better than that without equalization, with richer details, more natural color transition and better visual effect. This is also the case in literature [16]. From the column perspective, compared with [16], our model rarely shows color bleeding.

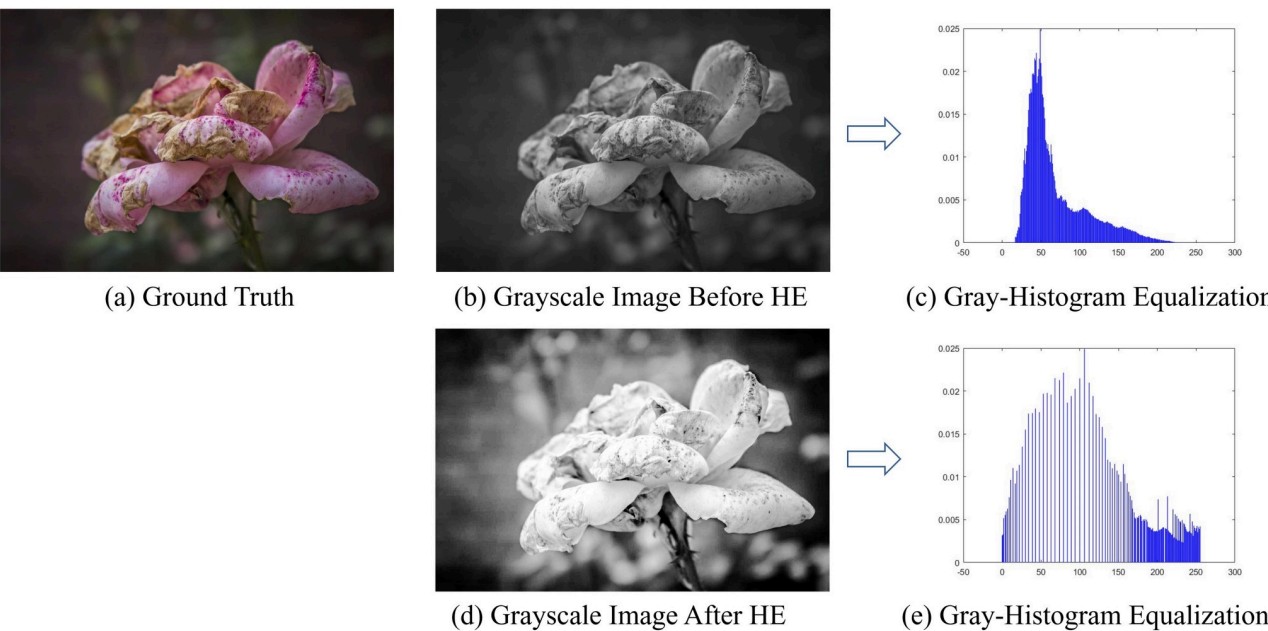

(a) Ground Truth  (b) Grayscale Image Before HE  (c) Gray-Histogram Equalization

(d) Grayscale Image After HE  (e) Gray-Histogram Equalization

**Fig 5. The image and its histogram before and after gray-histogram equalization.**

By analyzing the data in Tables 3 and 4, we found that the $C$ values of the colored image predicted by these model are mostly lower than the $C$ values of the ground truth. There are a few special cases, such as the img_5 in Table 3 and the img_3, img_4, img_5 in Table 4, the $C$ values of the image obtained after adding histogram equalization to our model and literature [16] are significantly higher than the $C$ values of the ground truth. Through the analysis of these data, it is not difficult to find that adding histogram equalization to the coloring model does not help the color prediction of normally exposed images, but may reduce the $C$ value of the image, while for the image with abnormal exposure, it can effectively increase the $C$ value of the predicted image. For img_1 and img_2 that are relatively normally exposed, whether to add histogram equalization has little effect on the $C$ values of the image, that is, $\Delta C_2$ is close to 0. For img_3, img_4, img_5 and img_6 that are not properly exposed, the model with histogram equalization added is significantly more accurate in predicting the color of grayscale images. These $C_{after}$ values of the colored image are higher or slightly different than these $C_{before}$ values of the image without histogram equalization in a large probability, that is, $\Delta C_2$ is much higher than 0 or close to 0.

**Comparison with state-of-the-art algorithms.** Fig 7 shows the comparison of the global coloring effects of our model and five classic models [16–18, 47, 48] in different complexity scenes. Through comparing the coloring effects of eight groups of test images, we find that the images processed by three models proposed in 2016 have obvious color bleeding, the images processed by Lei et al. [47] have simple colors and Su et al. [48] have slight color overflow, while our algorithm with the high-level semantic segmentation information of the image itself has strong robustness, which can apply to natural image colorization in different scenes.

In general, these algorithms are more accurate in dealing with highly recognizable scenes, while color bleeding and unclear edges may occur for difficult to recognize parts. In order to further verify the advantages of our algorithm, we invite 20 college students (10 women and 10 men, ranging from 20 to 30 years old) with normal vision, and ask them to score the coloring effects of these algorithms in terms of the three indexes given in Table 5. The test content is the

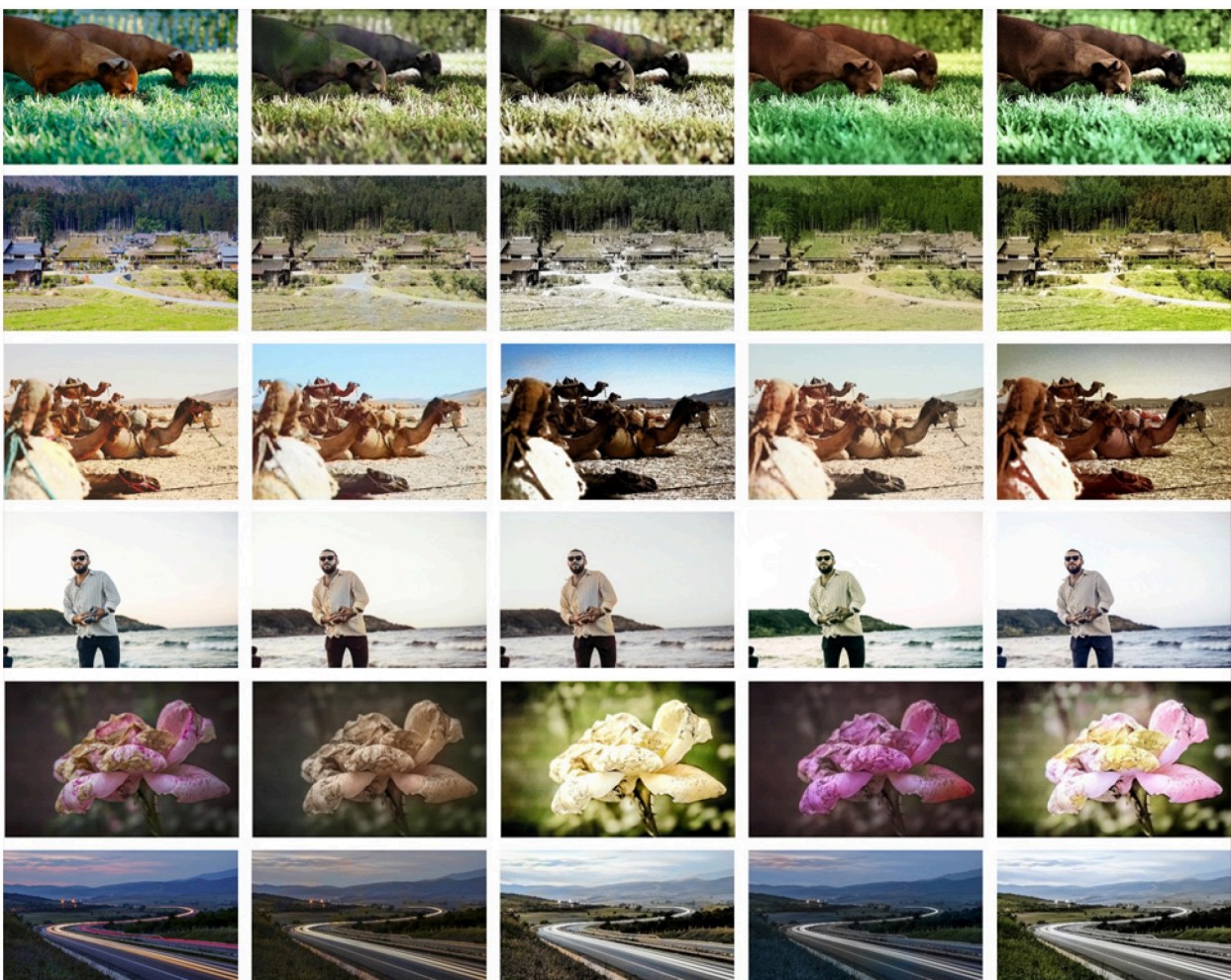

**Fig 6. The effect of HE on image colorization.** From top to bottom: two normal exposure image, two overexposed images, two underexposed images. From left to right: the ground truth, the coloring effect of Iizuka et al. [16] (the original method), the coloring effect of Iizuka et al. [16] with histogram equalization(the improved method), the coloring effect of ours without histogram equalization, and the coloring effect of ours with histogram equalization.(HE = histogram equalization). (a) GT, (b) [16], (c) [16] + HE, (d) Ours—HE, (e) Ours.

**Table 3. The influence of HE on the colorfulness of Iizuka et al. [16].**

| [16] | $C_{GT}$ | $C_{before}$ | $C_{after}$ | $\Delta C_1$ | $\Delta C_2$ |
|---|---|---|---|---|---|
| img_1 | 29.243 | 24.987 | 23.878 | -5.365 | -1.109 |
| img_2 | 31.036 | 15.735 | 13.246 | -17.790 | -2.489 |
| img_3 | 37.986 | 33.255 | 34.855 | -3.131 | 1.600 |
| img_4 | 6.572 | 9.351 | 9.438 | 2.866 | 0.087 |
| img_5 | 27.276 | 23.788 | 37.396 | 10.120 | 13.608 |
| img_6 | 19.610 | 11.929 | 12.790 | -6.820 | 0.861 |

$C_{GT}$ is the colorfulness of the ground truth. $C_{before}$ is the colorfulness of the colored image without HE. $C_{after}$ is the colorfulness of the colored image with HE. $\Delta C_1 = C_{after} - C_{GT}$, $\Delta C_2 = C_{after} - C_{before}$. The data in the table is from Fig 6, and the images are in the same order.

**Table 4. The influence of HE on the colorfulness of ours.**

| Ours-HE | $C_{GT}$ | $C_{before}$ | $C_{after}$ | $\Delta C_1$ | $\Delta C_2$ |
|---|---|---|---|---|---|
| img_1 | 29.243 | 26.682 | 26.512 | -2.731 | -0.170 |
| img_2 | 31.036 | 27.254 | 26.176 | 5.14 | -1.078 |
| img_3 | 37.986 | 31.476 | 39.692 | 1.706 | 8.216 |
| img_4 | 6.572 | 5.252 | 6.784 | 0.212 | 1.532 |
| img_5 | 27.276 | 35.011 | 30.977 | 3.701 | -4.034 |
| img_6 | 19.610 | 2.240 | 14.475 | -5.135 | 12.235 |

The meaning of each parameter in Table 4 is the same as that in Table 3. The data in the table is from Fig 6, and the images are in the same order.

above eight groups of images, the highest score in each group of images is 5, the lowest score is 1, the same score can appear in the same group of images. Then, we calculate the average score of each algorithm under these three indexes, and get the Table 5. After comparing these data in Table 5, it is found that ours has higher scores than other five algorithms in these three indexes. The comprehensive score is 4.27, which is at least 0.15 higher than the scores of other algorithms, which shows that the robustness of ours is very good, and the coloring effect in different scenes is relatively stable.

Tables 6–8 show the PSNR values, the SSIM values and the QSSIM values of the above eight groups of images in turn. The data marked in bold is the top three best values obtained by

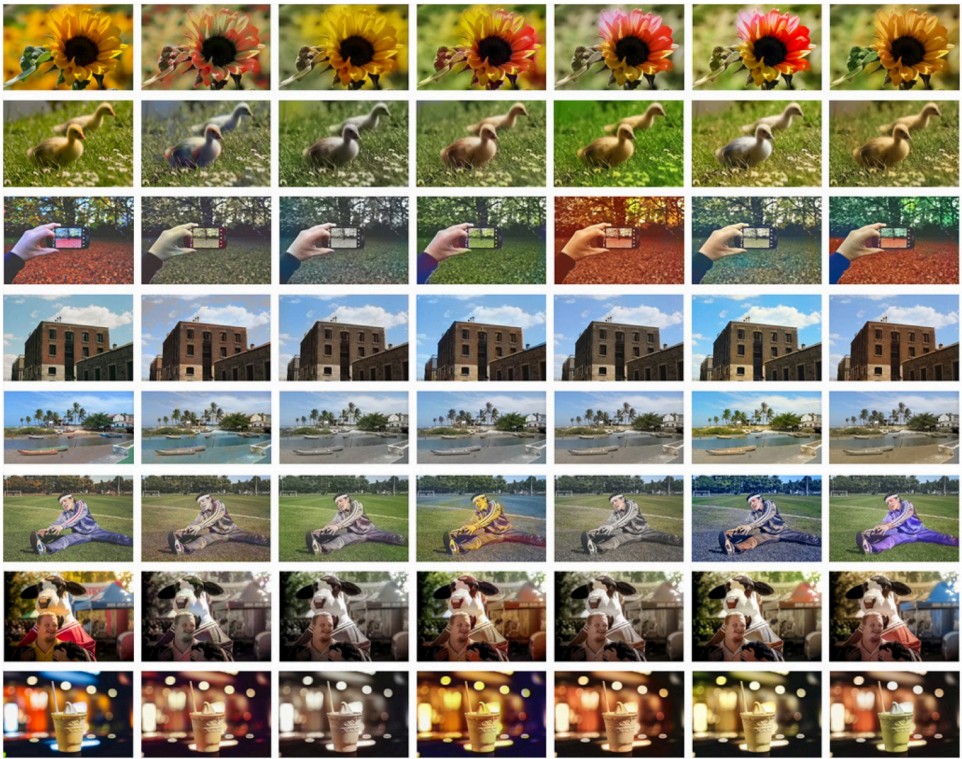

**Fig 7. Recolor of natural images.** The first four are simple natural scenes such as the lawn, single object, the sky and simple architecture, while the last four are complex natural scenes such as the water, many objects, brilliant lights and complex color levels. Literature [48] has two ways of automatic coloring and manual coloring, and the effect of automatic coloring is shown here. (a) GT, (b) [16], (c) [17], (d) [18], (e) [47], (f) [48], (g) Ours.

**Table 5. User ratings.** The data in the table is from Fig 7, and the images are in the same order.

| Methods | Index 1 | Index 2 | Index 3 | Composite Scores |
|---|---|---|---|---|
| [16] | 3.86 | 3.38 | 3.76 | 3.67 |
| [17] | 3.63 | 3.42 | 3.59 | 3.55 |
| [18] | 3.81 | 3.58 | 3.47 | 3.62 |
| [47] | 4.03 | 3.69 | 3.71 | 3.81 |
| [48] | 4.13 | 4.03 | 4.19 | 4.12 |
| Ours | 4.31 | 4.21 | 4.28 | 4.27 |

The three indexes are the rationality of coloring(Index 1), the naturalness of color transition(Index 2), the richness of color(Index 3).

**Table 6. PSNR.** The data in Tables 6–8 are all from Fig 7, and the order of images is the same.

| Methods | img_1 | img_2 | img_3 | img_4 | img_5 | img_6 | img_7 | img_8 | Average |
|---|---|---|---|---|---|---|---|---|---|
| [16] | **51.551** | **51.053** | **49.859** | **50.765** | **51.020** | **51.412** | **50.534** | **50.961** | **50.8944** |
| [17] | 28.721 | 27.795 | 32.348 | 36.392 | 36.192 | 35.279 | 31.759 | 30.926 | 32.4265 |
| [18] | **36.066** | **38.247** | **35.511** | 44.046 | **41.617** | 40.724 | 34.409 | 34.811 | **38.1789** |
| [47] | 32.294 | 32.121 | **36.407** | 49.702 | 41.471 | 42.675 | **34.493** | **35.656** | 38.1024 |
| [48] | 22.811 | 23.696 | 26.362 | 28.799 | 25.065 | 23.417 | 25.628 | 25.791 | 25.1961 |
| Ours | **37.786** | **39.326** | 34.390 | **49.291** | 41.152 | **42.714** | 34.918 | **34.966** | **39.3179** |

using different methods. In Table 6, literature [16] has obvious advantages, and the images processed by it all get the best PSNR value. However, the performance of our algorithm is also good. Among the eight PSNR values, the second place accounts for 1/2, the third place accounts for 1/4, and the fourth place also accounts for 1/4. In Table 7, The SSIM values of the images processed by these six algorithms are good and the differences between them are very small. However, our algorithm performed very well. Among the eight SSIM values, the first place accounts for 5/8, the second place accounts for 1/4, and the fourth place account for 1/8. In Table 8, the QSSIM values the images processed by the six methods are nice, and the differences between them are also small. Similarly, among the eight SSIM values of our algorithm, the first place accounts for 1/4, the second place accounts for 1/2, and the third and fourth place account for 1/8 respectively. Although the PSNR, SSIM and QSSIM values of color images predicted by our model are not always optimal, there is a small gap between them and the optimal values. At the same time, it can be seen from the obtained mean values that the objective indicators of these images obtained by our model are all good and can basically meet the requirements of users.

**Application effects of state-of-the-art algorithms on black-and-white images colorization.** Due to the long-term fading of these historical images and old photos, it is urgent to

**Table 7. SSIM.**

| Methods | img_1 | img_2 | img_3 | img_4 | img_5 | img_6 | img_7 | img_8 | Average |
|---|---|---|---|---|---|---|---|---|---|
| [16] | **0.897** | **0.844** | 0.889 | **0.947** | 0.860 | **0.874** | **0.958** | **0.963** | **0.9040** |
| [17] | **0.897** | **0.850** | **0.890** | 0.945 | **0.862** | 0.872 | 0.957 | 0.961 | **0.9043** |
| [18] | **0.898** | 0.839 | **0.890** | 0.946 | **0.861** | 0.873 | **0.958** | 0.962 | 0.9034 |
| [47] | 0.894 | **0.842** | **0.891** | **0.947** | **0.861** | **0.874** | **0.958** | **0.963** | 0.9038 |
| [48] | 0.850 | 0.830 | 0.873 | 0.944 | 0.853 | 0.841 | 0.942 | 0.950 | 0.8854 |
| Ours | **0.898** | 0.839 | **0.890** | **0.947** | **0.861** | **0.875** | **0.958** | **0.963** | **0.9039** |

**Table 8. QSSIM.**

| Methods | img_1 | img_2 | img_3 | img_4 | img_5 | img_6 | img_7 | img_8 | Average |
|---|---|---|---|---|---|---|---|---|---|
| [16] | 0.898 | **0.938** | 0.918 | 0.965 | 0.964 | **0.975** | 0.944 | 0.932 | 0.9418 |
| [17] | **0.934** | **0.955** | **0.924** | **0.980** | **0.973** | **0.979** | **0.948** | **0.946** | **0.9549** |
| [18] | 0.891 | 0.928 | 0.908 | 0.975 | 0.969 | 0.960 | 0.933 | 0.902 | 0.9333 |
| [47] | **0.912** | 0.926 | **0.924** | **0.984** | **0.973** | **0.975** | **0.950** | **0.943** | **0.9484** |
| [48] | 0.879 | 0.926 | **0.920** | 0.969 | 0.960 | 0.920 | 0.932 | 0.932 | 0.9298 |
| Ours | **0.934** | 0.936 | **0.920** | **0.983** | **0.972** | 0.972 | **0.949** | **0.947** | **0.9516** |

study a robust colorization algorithm to rescue them. Fig 8 shows the colorization effects of several algorithms on several groups of black-and-white images. At first glance, it is found that the colorization effects of these algorithms are very good. Compared with the original black-and-white image, the visual effect of the colored image has improved a lot. After magnification of the parts, it is found that in contrast to these methods, our model and literature [48] have a very uniform and stable effect on people's skin, clothing, plants, sky, natural light and so on. In complex scenes, such as the last two groups of images, the colors predicted by four models [16–18, 47] are not only single in color, but also appear a lot of colors overflow. However, the coloring effect of this paper and literature [48] are better, the color of the image is not only rich, but the transition between each other is very natural.

Boxplot data in Fig 9 comes from 20 testers who sort the six groups of images in Fig 8. Some data can be drawn from diagram: the effect generated by Iizuka et al. [16] has a probability distribution of 50% between 3 − 5, and its color effect is relatively stable, but not outstanding; both Larsson et al. [17] and Lei et al. [47] have a probability distribution of 50% between 2 − 5, 25% of them are between 2 − 4, and they are better than the former; Zhang et al. [18] have a probability distribution of 50% between 2 − 6, 25% of them are between 4 − 6, and its data is more scattered than the former; Su et al. [48] has better stability, with its 25% is distributed between 1 − 2, 50% is distributed between 2 − 4.5, and 25% is distributed between 4.5 − 6, which is better than the previous four algorithms; the images processed by our method, 25% of them is distributed between 1 − 2, 50% is distributed between 2 − 5, and 25% is distributed between 5 − 6, which is slightly lower than literature [48] in general. In the objective evaluation in the previous section, the advantages of our method are not particularly prominent, but the results of this survey show that the subjective evaluation effect of our method is better than the images colored by four models [16–18, 47], and it's more consistent with people's subjective perception, which will be more meaningful.

To highlight the advantages of our algorithm, we ask these 20 testers to do the third test. The scoring object includes 100 groups of images, each group has six images. These six images are in turn the coloring results of five classical algorithms [16–18, 47, 48] and ours. Each group of image is only displayed for ten seconds, and the testers must score and sort them immediately after reading a set of images. Unlike the previous two manual tests, this test requires the testers to directly sort and rank each group of images according to their own feelings at this time, and the results are shown in Table 9. Our model has the highest hit rate in the top 1 and the top 3. Among them, the top 1 is about 38%, which is far more than other algorithms, and the last hit rate is only 6%, which is much lower than other algorithms. The datas show that the coloring effects of our model are better than other algorithms in general.

**Limitations and transferability test of ours.**　Fig 10 shows the comparison of the color prediction effects of the six image colorization methods on the five groups of images. Several comparison cases show that the transferability of our algorithm is visually superior to the other

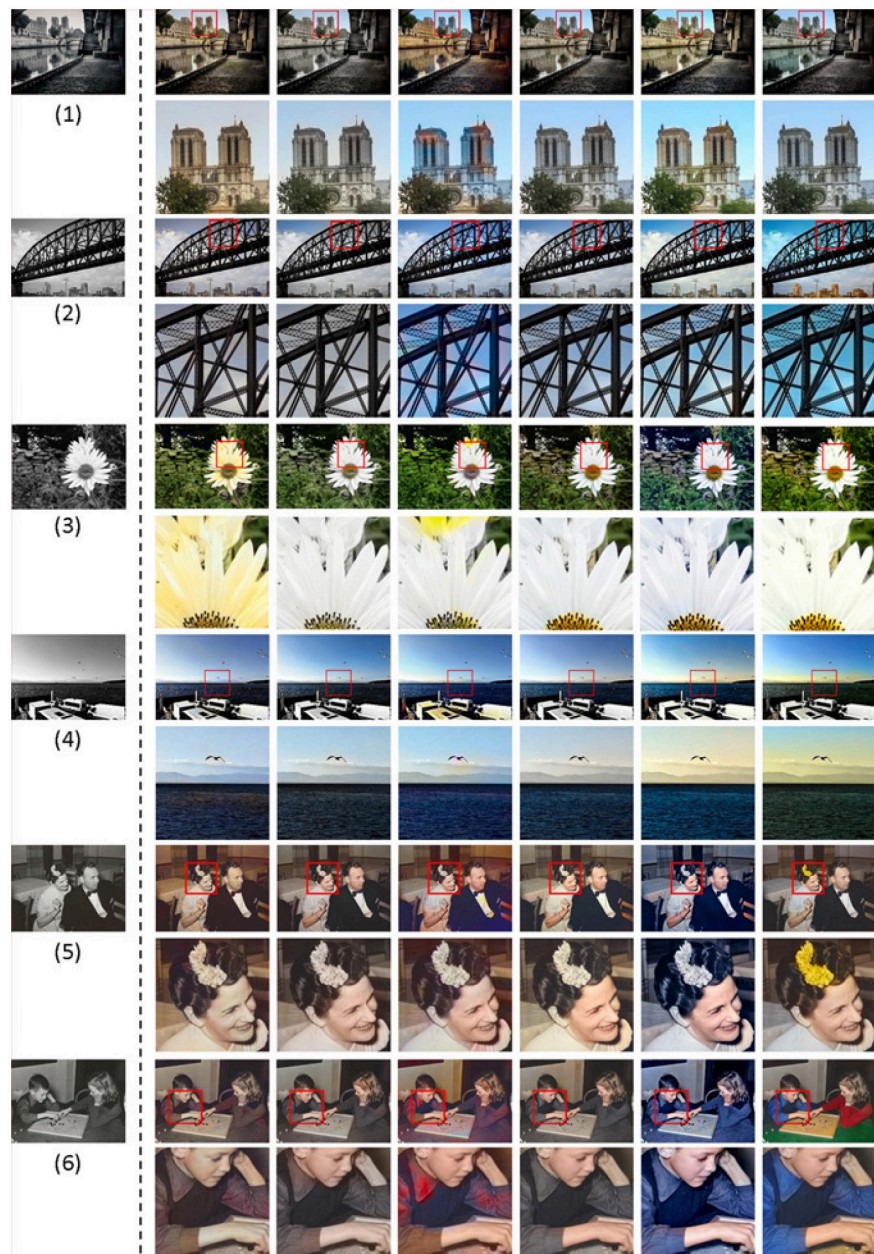

**Fig 8. Color restoration of black-and-white images.** (a) GT, (b) [16], (c) [17], (d) [18], (e) [47], (f) [48], (g) Ours.

five algorithms. The examples listed in this paper are limited, but these cases are also relatively common and many algorithms are not well handled. In most cases, the experimental results of our algorithm are almost the same as the ground truth, and the colors of the images are also very bright. Although there may be deviations from the ground truth, it is still semantically correct. Nowadays, colorization technology is no longer only used for black-and-white photos, but also has a widely fields included old movies, medical image, cartoon coloring, the restoration of cultural relics and artwork, statue restoration, remote sensing images and so on. Therefore, our algorithm will have a large application market.

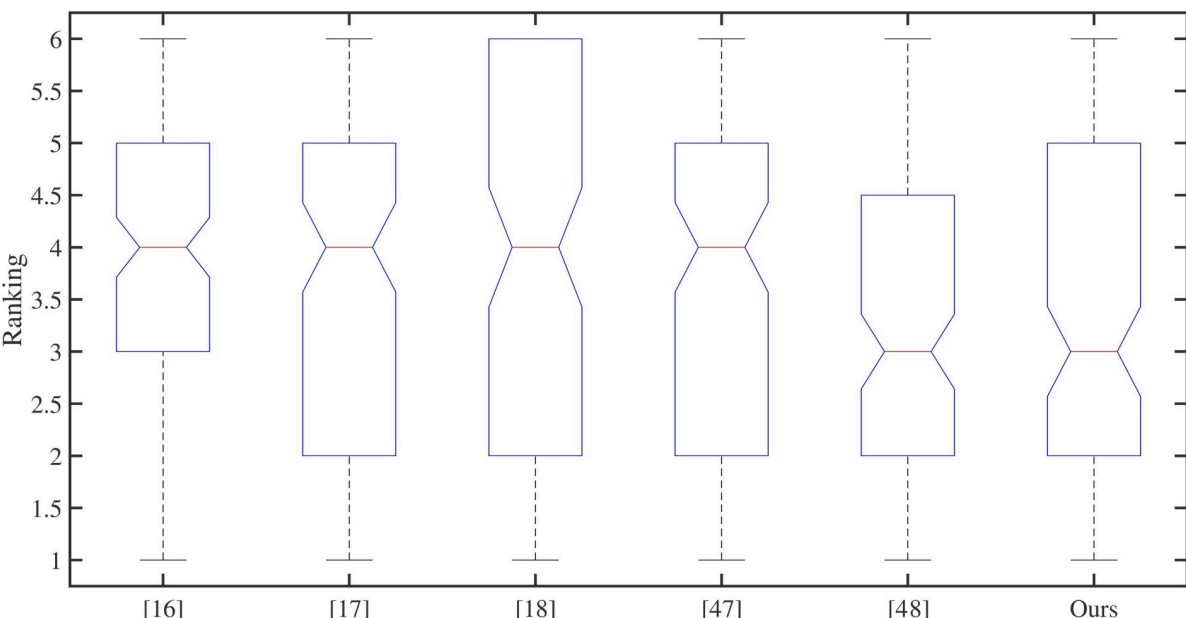

**Fig 9. Ranking distribution of coloring effects of six algorithms.** Each group of images contains the coloring results of six algorithms, which are sorted in ascending order of 1–6. No parallel ranking is allowed in the same group of images. Among them, the smaller the sorted number, the better the coloring effect processed by the algorithm.

Our model also has some areas that need to be improved. Fig 11 shows a few cases of poor coloring with our model. The following images contain too many targets, too dense targets, too small targets and no depth information, resulting in poor coloring effects, and problems such as warming the grayscale image directly, a mess of colors, blurry edges, and inactive colors. From the second row of images, we can see that our model is still very effective. For example, the red rose in the first column, white metal bookrack in the second column, white billboard in the crowd in the third column, dining table in the fourth column, basketball frame line between the sky in the fifth column, blond hair in the sixth column, white stripes in a striped socks in the seventh column, barbed wire in the eighth column, overall lighting atmosphere in the ninth column, which are endowed with the right color.

**Comparison of calculation speed of several algorithms.** Table 10 shows the average running time of these six colorization models when testing an image on the CPU and GPU respectively. We find that running code on the GPU is at least twice as fast as running code on the CPU. According to the data in Table 10, Lei et al. [47] takes the longest time to realize the

**Table 9. The ranking analysis of coloring effects of six algorithms.** Each set of images is still sorted in ascending order of $1 - 6$. The formula satisfied here is as follows: $Top1 = No.1$, $Top3 = No.1 + No.2 + No.3$, $Last1 = No.6$. It should be noted that the values of the last three columns in Table 9 are reserved only for the integer portion of the percentage.

| Methods | No.1(%) | No.2(%) | No.3(%) | No.4(%) | No.5(%) | No.6(%) | Top 1(%) | Top 3(%) | Last 1(%) |
|---|---|---|---|---|---|---|---|---|---|
| [16] | 9 | 9.9 | 20.05 | 13.5 | 21.85 | 25.7 | 9 | 39 | 26 |
| [17] | 9.6 | 15.6 | 14.2 | 23.2 | 17.25 | 20.15 | 10 | 39 | 20 |
| [18] | 12.35 | 11 | 20.2 | 24.25 | 15.2 | 17 | 12 | 44 | 17 |
| [47] | 15.95 | 25.9 | 13.55 | 17.3 | 14.85 | 12.45 | 16 | 55 | 12 |
| [48] | 15.35 | 24.1 | 11.2 | 14.05 | 16.95 | 18.35 | 15 | 51 | 18 |
| Ours | 37.75 | 13.5 | 20.8 | 7.7 | 13.9 | 6.35 | 38 | 72 | 6 |

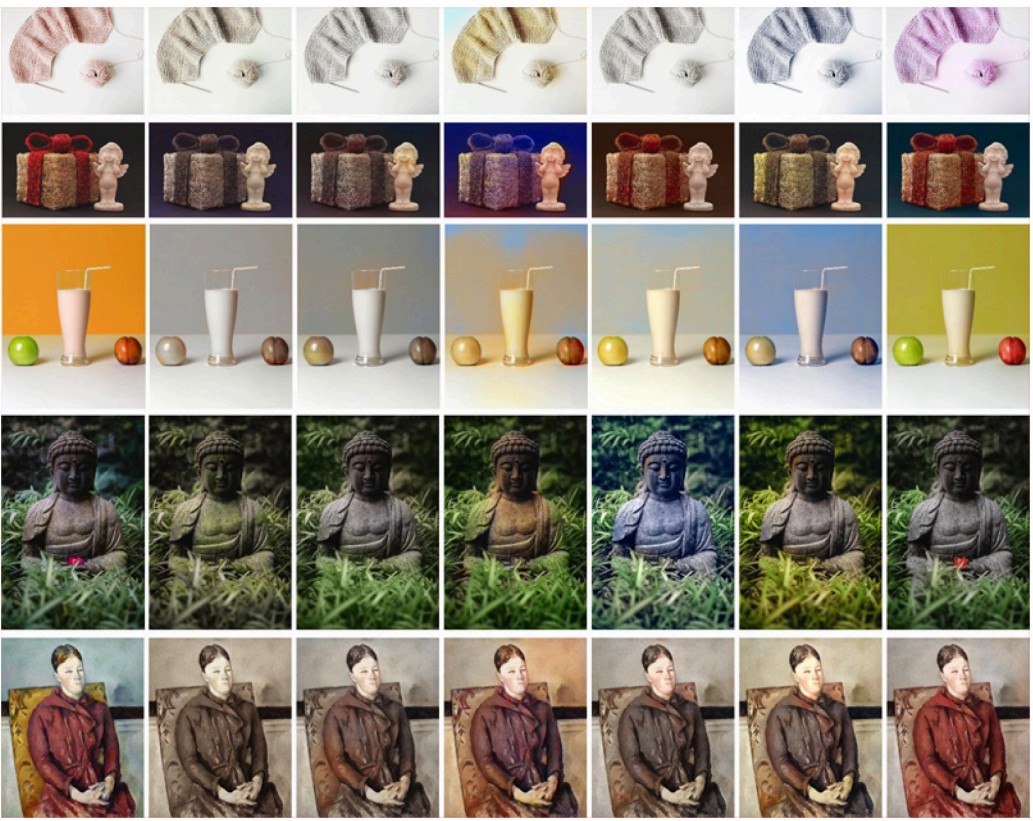

**Fig 10. Transferability test of our algorithm.** These five groups of images show the comparison of coloring effects of six algorithms on different materials, including wool, forging tape, ceramic, glass, stone Buddha, oil painting, etc. (a) GT, (b) [16], (c) [17], (d) [18], (e) [47], (f) [48], (g) Ours.

image colorization, whether it is running on the CPU or the GPU, and our model is the second, Zhang et al. [18] ranks the third, literature [16, 17, 48] spends the shortest time. However, if the performance of the algorithm is compared according to the computing time of GPU, our algorithm can complete this operation in 5 seconds, which is not far different from the running speed of literature [16–18, 48], which is also quite good.

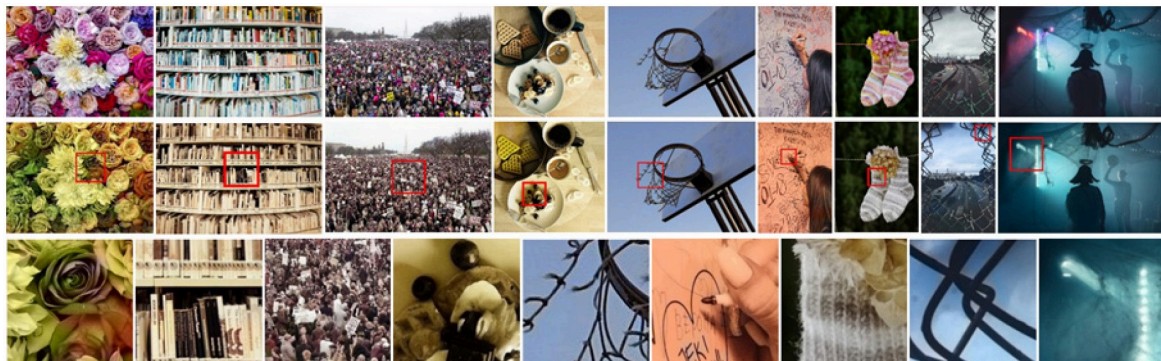

**Fig 11. Limitations of our algorithm.** The three rows of images from top to bottom are the ground truth, ours, and the partial zoom effect corresponding to the red box in the second row.

**Table 10. Comparison of calculation speed.** The data in Table 10 is the average running time of each image on the CPU/GPU, which is obtained by dividing the total time spent on testing the six models on the ILSVRC2012 by the total number of images.

| Methods | Num. lter. | CPU Time(s) | GPU Time(s) | Speedup |
|---|---|---|---|---|
| [16] | 1 | 4.576 | 1.015 | 4.508 |
| [17] | 1 | 4.262 | 1.036 | 4.114 |
| [18] | 1 | 6.173 | 2.452 | 2.518 |
| [47] | 1 | 25.281 | 7.602 | 3.326 |
| [48] | 1 | 4.236 | 2.054 | 2.062 |
| Ours | 1 | 11.826 | 3.806 | 3.107 |

## Conclusions

Since color images have incomparable advantages over black-and-white images in terms of people's visual perception and subsequent image understanding and analysis, it is of great significance to continue to study a practical grayscale image colorization algorithm. There are four advantages to our algorithm. Firstly, manual coloring requires high professional knowledge of users, and a little negligence will cause color matching problems. However, the biggest advantage of our model is automation, which does not require manual intervention and only requires the user to provide a target grayscale image. Secondly, our model can predict the two color layers $a^*$ and $b^*$ by using the gray information $L^*$ of the gray image itself as much as possible. Third, the network is able to capture and use semantic information, which makes the predicted color correct even if it is not close to the ground truth, which completely explains the problem that a single grayscale image may correspond to many reasonable color images. Fourth, we do pre-processing before the image is input to the network, which can effectively improve the color quality of overexposed and underexposed images, and increase the colorfulness of the image. In addition, our model can not only colorize grayscale images, but also colorize videos. Here, we only need to turn the video into a series of consecutive images before entering the network. As mentioned in the previous section, There are still some defects in our model, for example, our model has poor effect on this kind of image with many targets, small targets, dense targets and no depth information (see Fig 11). On top of that, there are also other limitations, for example, our model can neither generate strange color that formed by artists, nor automatically imagine the light, shade and complex texture in the comic manuscript. Therefore, it's necessary to rich the kinds of images of training set to enhance the generalization ability of neural network. In the following study, we will further improve the performance of the model and make the model learn people's visual aesthetics to color the image as much as possible.

## Supporting information

**S1 File. The data source for Fig 9.** These data were obtained by 20 testers who ranked the six groups of images in Fig 8 from $1 - 6$. Among them, this image is ranked in the first place represents the best colorization effect and this image is ranked in the sixth place represents the worst colorization effect. There are a total of 120 groups of data, and each group of data in turn corresponds to the subjective ranking of the processing effects of these six colorization algorithms.
(XLSX)

**S2 File. The source address download page for all images involved in Figs 1–11.** These pages contain the copyright holder and the copyright license information.
(XLSX)

**S1 Data. The data sources of Tables 5–9.** The Table 5 in the compressed package records the scores of 20 testers on eight groups of images in Fig 7 according to the given three indexes and shows the calculation process of the final composite scores of each algorithm. The Tables 6–8 show the PSNR values, the SSIM values and the QSSIM values of eight groups of images in Fig 7. The Table 9 records the coloring effect evaluation of 100 groups of images by 20 testers, and each group of images corresponds to the processing effect of six colorization algorithms in turn. Testers need to sort them from 1 – 6 according to their subjective consciousness.
(ZIP)

**S1 Text.**
(TXT)

## Acknowledgments

The authors are grateful to Dr. Bing Yu, a researcher who specializes in computer graphics and video image restoration, Shanghai Film Special Effects Engineering Technology Research Center; Zhihua Zheng, an IT worker who is interested in deep learning technologies and proficient in a variety of high-level language grammars, ICBC-AXA LIFE Insurance Company Limited; They gave us a lot of advice and ideas for writing.

## Author Contributions

**Conceptualization:** YouDong Ding.

**Data curation:** Min Xu.

**Formal analysis:** Min Xu.

**Funding acquisition:** YouDong Ding.

**Investigation:** Min Xu.

**Methodology:** Min Xu.

**Project administration:** Min Xu.

**Resources:** Min Xu.

**Software:** Min Xu.

**Supervision:** YouDong Ding.

**Validation:** Min Xu.

**Visualization:** Min Xu.

**Writing – original draft:** Min Xu.

**Writing – review & editing:** Min Xu.

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
