## [Decision Letter · Decision Letter 0]

12 Feb 2021

PONE-D-20-38461

Fully automatic image colorization based on semantic segmentation technology

PLOS ONE

Dear Dr. Ding,

Thank you for submitting your manuscript to PLOS ONE. After careful consideration, we feel that it has merit but does not fully meet PLOS ONE’s publication criteria as it currently stands. Therefore, we invite you to submit a revised version of the manuscript that addresses the points raised during the review process.

I found this manuscript well written and interesting.  Both reviewers were unanimous in their opinion that your manuscript describes technically sound piece of scientific research, however, they have made certain observations which need to be addressed. 

After thorough consideration of comments from all the reviewers, I felt that your study has merit but identified points that need to be addressed. Therefore, my decision is “major revision”.

We look forward to receiving your revised manuscript.

Kind regards,

Gulistan Raja

Academic Editor

PLOS ONE

Journal Requirements:

2. In your Methods section, please include additional information about your dataset, in particular the testing images, and how it was collected, in enough detail for another researcher to replicate the findings.

3.We note that the grant information you provided in the ‘Funding Information’ and ‘Financial Disclosure’ sections do not match.

4. We note that Figure(s) 2, 3, 5, 6, 7, 8 and 10 in your submission contain copyrighted images. All PLOS content is published under the Creative Commons Attribution License (CC BY 4.0), which means that the manuscript, images, and Supporting Information files will be freely available online, and any third party is permitted to access, download, copy, distribute, and use these materials in any way, even commercially, with proper attribution. For more information, see our copyright guidelines: http://journals.plos.org/plosone/s/licenses-and-copyright.

a) You may seek permission from the original copyright holder of Figure(s) 2, 3, 5, 6, 7, 8 and 10 to publish the content specifically under the CC BY 4.0 license.

Reviewers' comments:

Reviewer's Responses to Questions

**Comments to the Author**

1. Is the manuscript technically sound, and do the data support the conclusions?

Reviewer #1: Yes

Reviewer #2: Yes

2. Has the statistical analysis been performed appropriately and rigorously? 

Reviewer #1: Yes

Reviewer #2: Yes

3. Have the authors made all data underlying the findings in their manuscript fully available?

Reviewer #1: No

Reviewer #2: Yes

4. Is the manuscript presented in an intelligible fashion and written in standard English?

Reviewer #1: Yes

Reviewer #2: Yes

5. Review Comments to the Author

Reviewer #1: The subject of the paper is interesting and it fits well with the scope of the journal, and objectives are stated clearly.

I could not find any logical errors in the presentation and the approaches used. In the manuscript, it is peculiar that sometimes the authors use "I" if they refer to themselves, however the manuscript has two authors. The authors apply multiple performance indices to evaluate the quality of colorization, such as PSNR, SSIM, and MSE. However, MSE and PSNR does not indicate well the perceptual quality of digital images and SSIM is for grayscale images. I recommend the authors to use somekind of full-reference image quality metrics for RGB images, for example QSSIM ( https://searchcode.com/codesearch/view/85580087/ ). On the other hand, there are many full-reference image quality metrics for RGB images are available online (https://www.mdpi.com/1999-4893/13/12/313), so the authors can choose from many possible algorithms. The authors compared the proposed algorithm to several other state-of-the-art deep learning algorithms. It is not clear from the algorithm whether these methods were trained on the same database or not. I think a detailed description of the evaluation protocol would solve this problem. I think it is important to train all examined methods on the same dataset.

Reviewer #2: Aiming at these problems of image colorization algorithms based on deep learning, such

as color bleeding and insufficient color, the authors convert the study of image

colorization to the optimization of image semantic segmentation, and propose a fully

automatic image colorization model based on semantic segmentation technology. Although the idea and research questions of this paper are timely and clear, however, some major questions need to be addressed. My questions and comments are presented as follows that may be helpful to improve the quality of this paper.

1. The paper does not explain clearly its advantages with respect to the existing mehtods: it is not clear what is the novelty and contributions of the proposed work: does it propose a new method or any improvement of the existing models ? If so, please state clearly the modifications. There is slight novelty but there are several components of the framework that would require marked improvement.

2. In Table. 6 and Table .7, some PSNR and SSIM indexes are much higher than those in the compared methods. This issue further leads my following concerns about the generalization and overfitting in some occasions.

3. in discussing the complexity of your method, it might also be helpful to include an analysis of the empirical speed of different parts of the colorization process.

4. Could the techniques used or insights gained in this paper be applied to problems apart from

image colorization? If so, these could be discussed in the conclusion.

5. The literature has to be strongly updated with some relevant and recent papers focused on the fields dealt with the manuscript, such as "Automated colorization of a grayscale image with seed points propagation, IEEE Transactions on Multimedia 22 (7), 1756-1768,2020. Scene guided colorization using neural networks, Neural Computing and Applications, 2019."

6. PLOS authors have the option to publish the peer review history of their article (what does this mean?). If published, this will include your full peer review and any attached files.

Reviewer #1: No

Reviewer #2: No

---

## [Author Response · Author response to Decision Letter 0]

16 Sep 2021

Dear Editors and Reviewers，

Thank you for your letter and the reviewers’ comments on our manuscript entitled “Fully automatic image colorization based on semantic segmentation technology” (ID: PONE-D-20-38461). Those comments are all valuable and very helpful for revising and improving our paper, as well as the important guiding significance to our researches. We have carefully studied the opinions of the experts and actively made corrections, hoping to get approval. The test images used here are from the public dataset ILSVRC2012 and are freely available for academic research and non-commercial use. As the test images in the paper are replaced, the corresponding tabular data and textual analysis are adjusted accordingly. 

At the same time, the data materials attached to the original manuscript have been updated. In addition, we added a subsection "Comparison of Calculation Speed of Several Algorithms" at the end of the module of "Experimental Results and Analysis", which further enriched our evaluation indicators. Compared to the manuscript submitted in the first version, the revision we submitted this time shows significant changes. These comparisons can be viewed in the document labeled " Revised Manuscript with Track Changes ". 

We use the PDF manuscript file generated by the latex template provided on the official website of PLOS ONE, and then use Adobe Acrobat Pro to compare the old and new versions of the document and generate a "revised manuscript with track changes". What needs to be reminded is that for better viewing results, we recommend that experts use Adobe Acrobat Pro to open and view this "Revised Manuscript with Track Changes ". When you click on the highlighted text, you can also see the comparison of the changes between the old and the new versions. As shown in Fig 1:

Fig 1. Preview effect with modified track manuscript.

Responds to the Editor’s comments: 

Thanks for your comments on our paper. We have revised our paper according to your comments. If you have any question about this paper, please don’t hesitate to let me know. Here are our answers to each of these questions:

Comment 1：Please ensure that your manuscript meets PLOS ONE's style requirements, including those for file naming. The PLOS ONE style templates can be found at https://journals.plos.org/plosone/s/file?id=wjVg/PLOSOne_formatting_sample_main_body.pdf and https://journals.plos.org/plosone/s/file?id=ba62/PLOSOne_formatting_sample_title_authors_affiliations.pdf

Reply 1: Yes, we have confirmed that we are using the latest latex template (https://journals.plos.org/plosone/s/latex) provided on the official website of PLOS ONE, which should meet the style requirements of PLOS ONE. At the same time, we checked the naming of all files and corrected the corresponding errors, such as changing "fig1.eps" to "Fig1.eps". Images uploaded to the site should start with a capital letter “Fig.eps”.

Comment 2: In your Methods section, please include additional information about your dataset, in particular the testing images, and how it was collected, in enough detail for another researcher to replicate the findings.

Reply 2：We really appreciate your valuable advice. It is indeed our job to provide relevant information of the dataset. We are very sorry that because the training set and the test set used in the first manuscript submitted contain some images from the Internet, it is difficult for us to contact these original copyright holders, so we can only choose to delete most of the test images in this article, and change all to the images in ILSVRC2012. For an introduction to the training set and the test set used in the manuscript, please refer directly to page 8 lines 236-245. This paragraph introduces the training images, validation images, and test images used in this article and several comparison algorithms in the article, all from the same data set ILSVRC2012. The ILSVRC2012 dataset is the data set of the famous ImageNet2012 competition. Its download link is: http://www.image-net.org/challenges/LSVRC/2012/index.

Comment 3: We note that the grant information you provided in the ‘Funding Information’ and ‘Financial Disclosure’ sections do not match. When you resubmit, please ensure that you provide the correct grant numbers for the awards you received for your study in the ‘Funding Information’ section.

Reply 3：Financial Disclosure：This project was funded by National Natural Science Foundation of China under the grant 61303093 and 61402278. The funders had no role in study design, data collection and analysis, decision to publish, or preparation of the manuscript. None of the authors received salaries from any of the funders.

Comment 4: We note that Figure(s) 2, 3, 5, 6, 7, 8 and 10 in your submission contain copyrighted images. All PLOS content is published under the Creative Commons Attribution License (CC BY 4.0), which means that the manuscript, images, and supporting information files will be freely available online, and any third party is permitted to access, download, copy, distribute, and use these materials in any way, even commercially, with proper attribution. For more information, see our copyright guidelines: http://journals.plos.org/plosone/s/licenses-and-copyright.

Reply 4：Due to the difficulty of obtaining the permission from the original copyright holders to publish these data under CC BY 4.0, we have chosen to delete the original image and have replaced all the copyrighted images in Figures 2, 3, 5, 6, 7, 8 and 10 in the new manuscript. The images used in the present manuscript are from the ILSVRC2012 dataset, which was generated during the ImageNet Challenge in 2012. Each image in ImageNet belongs to the individual who provided the image. ImageNet does not own the copyright of the image. Subject to certain terms and conditions, we have obtained permission to use the ImageNet dataset for academic research and non-commercial purposes, This is shown in Fig 2.

Fig 2. A copyright certificate for ImageNet has been obtained.

Responds to the reviewer’s comments:

Reviewer #1:

The subject of the paper is interesting and it fits well with the scope of the journal, and objectives are stated clearly. I could not find any logical errors in the presentation and the approaches used. In the manuscript, it is peculiar that sometimes the authors use "I" if they refer to themselves, however the manuscript has two authors. The authors apply multiple performance indices to evaluate the quality of colorization, such as PSNR, SSIM, and MSE. However, MSE and PSNR does not indicate well the perceptual quality of digital images and SSIM is for grayscale images. I recommend the authors to use somekind of full-reference image quality metrics for RGB images, for example QSSIM ( https://searchcode.com/codesearch/view/85580087/ ). On the other hand, there are many full-reference image quality metrics for RGB images are available online (https://www.mdpi.com/1999-4893/13/12/313), so the authors can choose from many possible algorithms. The authors compared the proposed algorithm to several other state-of-the-art deep learning algorithms. It is not clear from the algorithm whether these methods were trained on the same database or not. I think a detailed description of the evaluation protocol would solve this problem. I think it is important to train all examined methods on the same dataset.

1. Response to comment: 

We are very grateful to you for your time and wisdom to provide suggestions for revisions to our articles to ensure that the highest-level articles are published. These suggestions really helped us a lot, for example, you proposed to use some kind of full reference image quality metrics for RGB images, so we specially added the new data, tables, pictures and other information about QSSIM in the revised manuscript. We have carefully studied your modification suggestions and summarized them into the following three questions (Comment 1.1, Comment 1.2, Comment 1.3). Below, we will also respond to these three questions in order (Reply 1.1, Reply 1.2, Reply 1.3) in a serious and responsible manner based on our research situation.

Comment 1.1: In the manuscript, it is peculiar that sometimes the authors use "I" if they refer to themselves, however the manuscript has two authors.

Reply 1.1：We are very sorry about the trouble caused by our wrong writing in Cover Letter and in the conclusion section (the sixth word in line 437 on page 14) of the first edition of the manuscript. You are absolutely right. Our manuscript has two authors. Whether it is in Cover Letter or in the manuscript, we should use it correctly. In our newly submitted Cover Letter and revised manuscript, we have carefully checked all the words and ensured that “I” no longer appear as the subject.

Comment 1.2: The authors apply multiple performance indices to evaluate the quality of colorization, such as PSNR, SSIM, and MSE. However, MSE and PSNR does not indicate well the perceptual quality of digital images and SSIM is for grayscale images. I recommend the authors to use some kind of full-reference image quality metrics for RGB images, for example QSSIM ( https://searchcode.com/codesearch/view/85580087/ ). On the other hand, there are many full-reference image quality metrics for RGB images are available online (https://www.mdpi.com/1999-4893/13/12/313), so the authors can choose from many possible algorithms.

Reply 1.2：As you said, MSE and PSNR are really not good indicators of the perceptual quality of digital images, so we also invited 20 volunteers to rate and sort the final results produced by using our model and five more advanced deep learning-based image colorization algorithms based on their own visual experience. The specific performance is as follows:

1) Volunteers were asked to rate the images processed by the six models from three indicators(the rationality of coloring(Index 1), the naturalness of color transition(Index 2), the richness of color(Index 3) according to their own intuitive feelings. For specific data, see “Tab5.xlsx” in the compressed package “S1_Data” in the attachment, which produced a total of 160 rows*18 columns of data. Then calculate the average score of each model in turn according to various indicators, and evaluate the performance of these methods in the perceptual quality of digital images. These contents are reflected in Table 5 of the revised manuscript(on page 12);

2) Volunteers were asked to rate and rank the overall quality of the coloring results of the same image processed by different methods according to their own intuitive experience, so as to evaluate the quality of each coloring method in terms of overall coloring effect based on these data. The specific data source can be seen in “Tab9.xlsx” in the compressed package “S1_Data” in the attachment (a total of 2000 rows * 6 columns of data are generated) and “S1_File.xlsx” (a total of 120 rows * 6 columns of data are generated). For the correspondence between the files in the attachment and the content in the manuscript, see the introduction of the module "Supporting information" (on page 16, lines 492-510). This part of the content is shown in Fig 9 and Table 9 of the revised manuscript(on page 14).

In the first version of the submitted manuscript, we combined PSNR, SSIM and Colorfulness as objective evaluation indicators, and user scoring as a subjective evaluation method to evaluate the quality of the colorized image. The content corresponding to this module can be found in a separate file labeled “Revised Manuscript with Track Changes” (on page 8, lines 260-262). SSIM is suitable for gray images, where we convert RGB to gray images before processing. PSNR, SSIM, QSSIM, etc. are all typical full-reference image quality indicators. We have also carefully read the information about QSSIM provided by you. We unanimously agree that it is reasonable to add QSSIM indicators for evaluation. In addition, the formula of QSSIM indicator is added to the module "Performance Evaluation Index" in the revised manuscript (on page 9, lines 286-290), and the analysis of QSSIM indicator is added to the module "Experimental Results and Analysis" (for example, Fig 4 on page 10 and Table 8 on page 13).

Comment 1.3: The authors compared the proposed algorithm to several other state-of-the-art deep learning algorithms. It is not clear from the algorithm whether these methods were trained on the same database or not. I think a detailed description of the evaluation protocol would solve this problem. I think it is important to train all examined methods on the same dataset.

Reply 1.3：Yes, the algorithm proposed in this article and several other most advanced deep learning-based image colorization algorithms are trained and tested on the same database, and the results are compared. This is described in the section "Experimental Environment and Dataset" of the revised manuscript (on page 8, lines 249-251).

Reviewer #2: 

Aiming at these problems of image colorization algorithms based on deep learning, such as color bleeding and insufficient color, the authors convert the study of image colorization to the optimization of image semantic segmentation, and propose a fully automatic image colorization model based on semantic segmentation technology. Although the idea and research questions of this paper are timely and clear, however, some major questions need to be addressed. My questions and comments are presented as follows that may be helpful to improve the quality of this paper.

2. Response to comment: 

Thank you very much for the excellent and professional revision of our manuscript. We found the reviewers’ comments to be helpful in revising the manuscript and have carefully considered and responded to each suggestion. In the majority of cases we were successful in incorporating the reviewers’ feedback into our revised manuscript. There were some very good suggestions that we couldn't complete in a short time, but we learned a lot from them.By answering the following questions, we know that there is still a lot to learn in this field. We will keep your ideas in mind and continue to do further research in this field.

Comment 2.1: The paper does not explain clearly its advantages with respect to the existing mehtods: it is not clear what is the novelty and contributions of the proposed work: does it propose a new method or any improvement of the existing models ? If so, please state clearly the modifications. There is slight novelty but there are several components of the framework that would require marked improvement.

Reply 2.1：Our network draws on the work of Iizuka et al. [16], and preprocesses the image before sending it to the network to improve the contrast of abnormally exposed images. At the same time, we add a semantic segmentation network to the model to optimize the coloring effect. These descriptions can be found in the manuscript (on page 3, lines 113-115). In addition, when designing the loss function, we considered the loss of the semantic segmentation network and the loss of the color prediction network(on lines 227 to 243 on pages 7 to 8 of the manuscript). The experimental results prove that the coloring effect obtained by using our model is better than the coloring effect of Iizuka et al. [16] in terms of objective indicators and subjective evaluation. Each set of experiments in this paper contains a comparison of the color effect of the proposed algorithm and this algorithm. The details can be seen from the beginning of line 296 on page 9 of the manuscript to the end of the experimental results and analysis. 

Compared with the existing methods, the fully automatic image colorization based on semantic segmentation technology method proposed in this article Its novelty is reflected in: (1) the addition of histogram equalization preprocessing for colored images, which makes our model have obvious advantages over other methods when dealing with over-dark or over-exposed images; (2) the introduction of semantic segmentation technology improves the accuracy of the algorithm and solves the problem of color overflow.

The contributions of this paper include(See lines 38-43 on page 2 of the manuscript): (1) histogram equalization effectively improves the visual effect and the colorfulness of overexposed and underexposed images;(2) the introduction of semantic segmentation network accelerates the edge convergence of the image and improves the positioning accuracy of the algorithm, and solves the problem of color bleeding; (3) compared with several popular algorithms, our model has better results in natural images colorization and old black-and-white images colorization. 

As for the several components you mentioned that need marked improvement, we have been thinking about it since we received this suggestion. Although we try to do some parameter modification and component innovation, the current results are not very good. It is very important. Because of your suggestions, we have discovered the shortcomings in our current work. We will follow your suggestions to improve the level of scientific research and achieve more results in future work. We are very happy to receive your guidance, and especially hope that this article can be accepted. 

Comment 2.2: In Table. 6 and Table .7, some PSNR and SSIM indexes are much higher than those in the compared methods. This issue further leads my following concerns about the generalization and overfitting in some occasions.

Reply 2.2：The question you raised is also what we thought about when designing the network model and designing the loss function. At that time, we also made preparations. If the model has a small error on the training set, but a large error on the test set, our solutions are two: Our solutions are two: (1) at the data level, you can use a simple matlab code to move, zoom, rotate, invert, and add noise to each image to obtain more data, thereby Realize data augmentation; (2) we can add L1 regularization to the loss function to prevent overfitting and improve generalization ability. At present, our model has not exhibited under-fitting and over-fitting. It is used in common natural image colorization (from page 11, line 354 to page 13, line 387), old black-and-white image colorization (from page 13, line 388 to page 14 line 429) and coloring of images containing special materials (from Page 15 line 430 to page 15 line 442) performed very well. Our model also has some areas that need to be improved (from page 15, line 443 to page 15, line 454). In the future, we will still be engaged in the colorization of images and videos. At the same time, we will continue to pay attention to this problem and continue to optimize the network structure.

Comment 2.3: in discussing the complexity of your method, it might also be helpful to include an analysis of the empirical speed of different parts of the colorization process.

Reply 2.3：Thank you for mentioning the analysis of operating speed. This is indeed our lack of consideration. We have also added an experiment to compare our own method with several other methods in terms of execution efficiency difference (on page 15, lines 455-464). However, we did not specifically discuss the execution efficiency of different parts of the colorization process. There are two main reasons: (1) although it is very good to be able to analyze the speed of different parts of the colorization process, as a complete automatic colorization method, the operating efficiency of a certain part of it cannot determine the final execution effect; (2) this time we are discussing the performance of our proposed method in completing the task of image colorization. It is not the focus of this article to look at the local running speed from the inside of the method. But if we want to continue to improve the running speed of our model in the future, it is natural to study the optimization space of different parts of the colorization process.

Comment 2.4: Could the techniques used or insights gained in this paper be applied to problems apart from image colorization? If so, these could be discussed in the conclusion.

Reply 2.4：This paper does not use any new technology, in which the semantic segmentation technology used to divide the image into different regions and extract the region of interest can be used in the fields of image classification, automatic driving of object detection, etc. However, this is already a well-known conclusion, which is not discussed in our conclusion section.

Comment 2.5: The literature has to be strongly updated with some relevant and recent papers focused on the fields dealt with the manuscript, such as "Automated colorization of a grayscale image with seed points propagation, IEEE Transactions on Multimedia 22 (7), 1756-1768,2020. Scene guided colorization using neural networks, Neural Computing and Applications, 2019."

Reply 2.5：Yes, we carefully studied the two articles you recommended, and they each proposed a fully automatic coloring algorithm, which is the same as our method. Wan S et al.[47] mentioned the subsequent application of the model to low-light night vision images in the conclusion part, and we pointed out in the contribution that the equalization can improve the coloring effect of low-exposure images (on page 2 lines 38-43). The specific experimental comparison effect can be seen in Fig 6 (on page 11). In the follow-up, we will also consider further expanding its scope of application. For example, applied to the color enhancement of low-light night vision images, and infrared image colorization.

Yu X et al. [48] used scene-guided coloring to essentially classify the coloring images first, and then quickly and effectively color them according to the characteristics of the images in the category. This makes coloring more targeted and reasonable in theory, and indeed has great reference significance , which we have quoted in the manuscript respectively(on page 3, lines 100-111). But in the experimental part, we did not compare them with the colorization effect of the algorithm in this paper. In the future research, we will pay attention to the update of the literature.

Among the five contrast algorithms, three are from the very classic grayscale image colorization algorithm in 2016[16,17,18]. At present, the latest research will generally take them as the contrast algorithm, because their coloring effect is really very excellent. The other two are in 2019[45] and 2020[46]. Among them, the paper in 2020 has been put into commercial application of PS Element. It should be noted that we chose its automatic coloring effect of the literature [46] in the comparative experiment, because its manual coloring effect is indeed very good, and it can be colored according to the user's own wishes. But this requires the user to have a certain color matching foundation, and the final product will be excellent.

The number of references referred to in the reply to this question corresponds to the order in the manuscript, as follows :

[16] Iizuka, Satoshi and Simo-Serra, Edgar and Ishikawa, Hiroshi. Let there be color!: joint end-to-end learning of global and local image priors for automatic image colorization with simultaneous classification. ACM Transactions on Graphics(TOG). 2016; 35(4):110.

[17] Larsson, Gustav and Maire, Michael and Shakhnarovich, Gregory. Learning representations for automatic colorization. European Conference on Computer Vision. Springer, 2016; pp.577-593.

[18] Zhang, Richard and Isola, Phillip and Efros, Alexei A. Colorful Image Colorization. arXiv preprint arXiv:1603.08511.2016.

[45] Lei, Chenyang and Chen, Qifeng. Fully Automatic Video Colorization With Self-Regularization and Diversity. Proceedings of the IEEE Conference on Computer Vision and Pattern Recognition. 2019; pp.3753-3761.

[46] Su, Jheng-Wei and Chu, Hung-Kuo and Huang, Jia-Bin. Instance-aware Image Colorization. IEEE Conference on Computer Vision and Pattern Recognition(CVPR). 2020.

[47] Shaohua Wan, Yu Xia, Lianyong Qi, Yee-Hong Yang, Mohammed Atiquzzaman. Automated Colorization of a Grayscale Image With Seed Points Propagation. IEEE Transactions on Multimedia. 2020, 22(7): 1756-1768.

[48] Xia, Y., Qu, S. & Wan, S. Scene guided colorization using neural networks. Neural Comput & Applic , 2018. https://doi.org/10.1007/s00521-018-3828-z.

We tried our best to improve the manuscript and made some changes in the manuscript. These changes will not influence the content and framework of the paper. Here we did not list all the modifications one by one, but made detailed replies according to the comments raised by academic editors and two reviewers, and pointed out some modifications related to these comments. See a separate file labeled “Revised Manuscript with Track Changes” for details of modifications. We appreciate for Editors/Reviewers’ warm work earnestly, and hope that the correction will meet with approval.

Once again, thank you very much for your comments and suggestions.

Sincerely yours,

---

## [Decision Letter · Decision Letter 1]

2 Nov 2021

Fully automatic image colorization based on semantic segmentation technology

PONE-D-20-38461R1

Dear Dr. Ding,

We’re pleased to inform you that your manuscript has been judged scientifically suitable for publication and will be formally accepted for publication once it meets all outstanding technical requirements.

Kind regards,

Gulistan Raja

Academic Editor

PLOS ONE

Additional Editor Comments (optional):

Reviewers' comments:

Reviewer's Responses to Questions

**Comments to the Author**

1. If the authors have adequately addressed your comments raised in a previous round of review and you feel that this manuscript is now acceptable for publication, you may indicate that here to bypass the “Comments to the Author” section, enter your conflict of interest statement in the “Confidential to Editor” section, and submit your "Accept" recommendation.

Reviewer #1: All comments have been addressed

Reviewer #2: All comments have been addressed

2. Is the manuscript technically sound, and do the data support the conclusions?

Reviewer #1: Yes

Reviewer #2: Yes

3. Has the statistical analysis been performed appropriately and rigorously? 

Reviewer #1: Yes

Reviewer #2: Yes

4. Have the authors made all data underlying the findings in their manuscript fully available?

Reviewer #1: Yes

Reviewer #2: Yes

5. Is the manuscript presented in an intelligible fashion and written in standard English?

Reviewer #1: Yes

Reviewer #2: Yes

6. Review Comments to the Author

Reviewer #1: The authors addressed all my comments very carefully. I was not able to find any logical errors in the presentation. I can recommend this manuscript for publication.

Reviewer #2: The authors have made significant progress in improving the paper.All issues raised by the concerns are addressed reasonably, so I am in favor of publication of the paper.

7. PLOS authors have the option to publish the peer review history of their article (what does this mean?). If published, this will include your full peer review and any attached files.

Reviewer #1: No

Reviewer #2: No

---

## [Editor Report · Acceptance letter]

12 Nov 2021

PONE-D-20-38461R1 

Fully automatic image colorization based on semantic segmentation technology 

Dear Dr. Ding:

I'm pleased to inform you that your manuscript has been deemed suitable for publication in PLOS ONE. Congratulations! Your manuscript is now with our production department. 

Kind regards, 

on behalf of

Dr. Gulistan Raja 

Academic Editor

PLOS ONE